# Language Generation with Replay:
# A Learning-Theoretic View of Model Collapse

**Giorgio Racca** [1]  **Michal Valko** [2]  **Amartya Sanyal** [1]

## Abstract

As scaling laws push the training of frontier large language models (LLMs) toward ever-growing data requirements, training pipelines are approaching a regime where much of the publicly available online text may be consumed. At the same time, widespread LLM usage increases the volume of machine-generated content on the web; together, these trends raise the likelihood of generated text re-entering future training corpora, increasing the associated risk of performance degradation often called *model collapse*. In practice, model developers address this concern through data cleaning, watermarking, synthetic-data policies, or, in some cases, blissful ignorance. However, the problem of model collapse in generative models has not been examined from a learning-theoretic perspective: we study it through the theoretical lens of the *language generation in the limit* framework, introducing a *replay* adversary that augments the example stream with the generator's own past outputs. Our main contribution is a fine-grained learning-theoretic characterization of when replay fundamentally limits generation: while replay is benign for the strongest notion of uniform generation, it provably creates separations for the weaker notions of non-uniform generation and generation in the limit. Interestingly, our positive results mirror heuristics widely used in practice, such as data cleaning, watermarking, and output filtering, while our separations show when these ideas can fail.

## 1. Introduction

Large language models (LLMs) are increasingly trained on web-scale corpora containing significant volumes of machine-generated text. As this fraction grows, a central concern is *model collapse* (Shumailov et al., 2024): the degradation of future models due to training on the outputs of their predecessors, effectively inflating the token count without adding new knowledge. While empirical evidence for such harmful feedback is accumulating, a principled theoretical understanding of when such feedback *fundamentally* limits the ability to generate language remains lacking.

We address this by building on the framework of *language generation in the limit* (Kleinberg & Mullainathan, 2024). Inspired by the classical literature on language identification (Gold, 1967; Angluin, 1980), this framework abstracts away specific language model architectures and training algorithms, and studies generation as an interactive game. An adversary first secretly selects a target language from a known class and then reveals an adversarially ordered stream of valid examples from that language; the generator is required to eventually produce an infinite sequence of previously unseen elements from the target language.

In this work, we propose a replay variant of the generation game, *language generation with replay*, that provides a minimal abstraction of the feedback loop underlying model collapse. In addition to valid examples from the target language, the adversary may inject the generator's own *previous outputs* into the example stream. This *replay* mechanism models synthetic content re-entering the data stream, a phenomenon that has been shown to be responsible for model collapse (Shumailov et al., 2024).

Recent work (Dmitriev et al., 2026) highlights the theoretical challenges posed by replay in online learning: a replay adversary can systematically mislead classical online learning algorithms, leading to strong separations between classical online learnability and online learnability under replay. We ask whether replay has an analogous effect on language generation.

> **Question.** Does the presence of replay, where a generator is trained on its own past outputs, make language generation fundamentally harder?

We answer this question with a fine-grained characterization across the main notions of *generatability*. The main

[1]University of Copenhagen [2]Isara Labs. Correspondence to: Giorgio Racca <g.racca@di.ku.dk>.

*Proceedings of the 43rd International Conference on Machine Learning*, Seoul, South Korea. PMLR 306, 2026. Copyright 2026 by the author(s).

Table 1. When is generatability unaffected by replay?

| Generation notion | Finite $\mathcal{H}$ | Countable $\mathcal{H}$ | General $\mathcal{H}$ |
|---|---|---|---|
| Uniform | ✓ (4.1) | ✓ (4.1) | ✓ (4.1) |
| Non-uniform | ✓ (4.1) | ✗ (5.1) | ✗ (5.1) |
| In the limit | ✓ (6.1) | ✓ (6.1) | ✗ (6.2) |
| Proper in the limit | ✗ (7.2) | ✗ (7.2) | ✗ (7.2) |

✓: same guarantees as the standard setting. ✗: strict separation from the standard setting. Parenthesized numbers indicate the theorem establishing the entry.

takeaway is that the effect of replay depends in a non-trivial way on the specific generation guarantee desired and on the complexity of the hypothesis class. Table 1 summarizes our main results. When replay does not affect generatability, we provide algorithms matching the guarantees of the standard setting; when it does, we construct hard instances that witness the separation.

The remainder of the paper is organized as follows. Section 2 discusses related work, while Section 3 introduces the replay model and states our contributions. Section 4 shows that uniform generation is equivalent in the standard and the replay model. Section 5 and Section 6 show that replay creates strict separations for non-uniform generation and generation in the limit, respectively, while also identifying regimes where these separations disappear. Section 7 shows that replay is even more restrictive for proper generation, where a separation already holds for finite classes. Finally, Section 8 discusses implications and open questions.

## 2. Related Work

**Generation in the limit.** Kleinberg & Mullainathan (2024) showed that *all* countable classes are generatable in the limit, whereas *identification* in the limit is only possible under restrictive assumptions (Gold, 1967; Angluin, 1980). This surprising separation sparked a flurry of follow-up work. One important line of research focuses on contrasting the notion of generatability with that of learnability (Raman et al., 2025; Bai et al., 2026; Hanneke et al., 2025); in particular, Raman et al. (2025) provided a characterization of uniform and non-uniform generatability via a novel combinatorial dimension. Another prominent line of follow-up work investigates the trade-off between avoiding hallucinations and maintaining output variety, as measured by "breadth" (Charikar & Pabbaraju, 2025; Kalavasis et al., 2025; 2026), "density" (Kleinberg & Wei, 2025), or being "representative" (Peale et al., 2025).

**Robust generation in the limit.** The line of research most closely related to ours extends the framework of language generation in the limit to allow *contaminations* in the exam-

ple stream, in the form of incorrect examples and omissions. Earlier work on generation from noisy examples focuses on the setting where the adversary is allowed to insert a *finite* number of *arbitrarily noisy* examples (Raman & Raman, 2025; Bai et al., 2026). Our setting is, in some sense, stronger, as we allow for an *infinite* number of noisy examples, and at the same time weaker, as we restrict the noisy examples to be among *previous outputs* of the generator. Recently, Mehrotra et al. (2026) studied the robustness of dense and non-dense generation in the limit under different regimes of *infinite* contamination. Our work differs from theirs in that the contamination rate is endogenously determined by the generator.

**Replay adversary in online learning.** In the online learning setting, Dmitriev et al. (2026) considered a similar *replay* adversary that can use previously output hypotheses to provide noisy labels. They showed that replay induces a separation from standard online learning and introduced a combinatorial dimension characterizing learnability in the replay setting. Their work can also be viewed as an attempt to formalize model collapse through the lens of learning theory. However, they focus on a supervised setting, while our analysis is tailored to the task of generation.

**Model collapse.** A growing body of research, often grouped under the umbrella term *model collapse* (Shumailov et al., 2024), examines the risks of recursively training models on the outputs of earlier generations, showing that this feedback loop can cause the tails of the distribution to be forgotten. Although the extent of this effect and its inevitability are subject to an ongoing debate, the evidence overall suggests that, in the absence of an adequate supply of high-quality data, model performance gradually deteriorates (Shumailov et al., 2023; Martínez et al., 2023; Briesch et al., 2023; Alemohammad et al., 2024; Bertrand et al., 2024; Gerstgrasser et al., 2024; Seddik et al., 2024; Zhang et al., 2024; Dohmatob et al., 2024; Marchi et al., 2024; Suresh et al., 2024; Dohmatob et al., 2025; Bohacek & Farid, 2025).

## 3. Setup and Results

This section formalizes the *language generation with replay* framework and informally states our main results.

### 3.1. Problem Setup

Let $\mathcal{X}$ be a countable domain and let $\mathcal{H} \subseteq \{0,1\}^{\mathcal{X}}$ be a binary hypothesis class. For $h \in \mathcal{H}$, write $\mathrm{supp}(h) := \{x \in \mathcal{X} : h(x) = 1\}$. As a concrete instantiation, the domain $\mathcal{X}$ may be taken to be the set of all tokens, or of all finite strings over a finite alphabet, with each hypothesis $h$ representing the language consisting of the strings in

supp $(h)$. Throughout, we assume supp $(h)$ is infinite for every $h \in \mathcal{H}$, since the goal is to output infinitely many distinct valid elements. This is referred to as the *uniformly unbounded support* (UUS) property by Raman et al. (2025).

Our starting point is the language generation game introduced by Kleinberg & Mullainathan (2024). The game proceeds over infinitely many rounds between an adversary and a (possibly computationally unbounded) *generator*, defined as a function $\mathcal{G}$ that maps each finite sequence $x_{1:t} \coloneqq (x_1, \ldots, x_t) \in \mathcal{X}^t$ to an output $o_t \coloneqq \mathcal{G}(x_{1:t}) \in \mathcal{X}$. At the start of the game, the adversary fixes a hidden target $h^\star \in \mathcal{H}$. In each round $t$, the adversary reveals an example $x_t \in$ supp $(h^\star)$, and the generator outputs $o_t = \mathcal{G}(x_{1:t})$. The generator *succeeds* if there exists a time $t^\star \in \mathbb{N}$ such that for all $t \geq t^\star$,

$$o_t \in \text{supp}(h^\star) \setminus \{x_1, \ldots, x_t\}.$$

No restriction is imposed on outputs before $t^\star$.

Different notions of generatability arise by varying what the success time $t^\star$ is allowed to depend upon. Under *uniform* generatability (Kleinberg & Mullainathan, 2024), $t^\star$ must be fixed across all $h \in \mathcal{H}$; under *non-uniform* generatability (Raman et al., 2025), it may depend on the particular target hypothesis $h^\star$; and under generatability *in the limit* (Kleinberg & Mullainathan, 2024), it may further depend on the specific sequence $(x_t)_{t \geq 1}$. For reference, relevant definitions and results from the standard setting are summarized in Appendix A; in particular, Table 2 contrasts the above three notions of generatability.

In this work, we introduce a minimal modification to the standard language generation game to capture the recursive feedback dynamics at the core of model collapse. While in the standard setting the adversary must reveal $x_t \in$ supp $(h^\star)$ in every round, in our replay setting the adversary may also reveal previous generator outputs, potentially including hallucinated ones. We refer to this variant of the standard language generation game as *language generation with replay*.

---

**Language Generation with Replay**

**Setup.** Hypothesis class $\mathcal{H} \subseteq \{0, 1\}^{\mathcal{X}}$ satisfying UUS.

**Game.** The adversary secretly picks $h^\star \in \mathcal{H}$.

For $t = 1, 2, \ldots$

  (i) The adversary reveals an example $x_t$ such that

    $x_t \in$ supp $(h^\star) \cup \{o_s \colon s < t\}$.

  (ii) The generator outputs $o_t \in \mathcal{X}$.

**Success.** There exists a finite time $t^\star \in \mathbb{N}$ such that $o_t \in$ supp $(h^\star) \setminus \{x_1, \ldots, x_t\}$ for all $t \geq t^\star$.

---

In the next section, we formally define several notions of

what it means for a class $\mathcal{H}$ to be *generatable with replay* and state our main results for each such notion.

## 3.2. Main Definitions and Results

The following definition formalizes the notion of adversarial sequences of examples *with replay*.

**Definition 3.1** (Sequence with replay for a hypothesis and a generator)**.** Fix a hypothesis $h$ and a generator $\mathcal{G}$. An infinite sequence $(x_t)_{t \geq 1}$ is a *replay sequence for $h$ and $\mathcal{G}$* if, for every $t \geq 1$,

$$x_t \in \text{supp}(h) \quad \text{or} \quad x_t \in \{\mathcal{G}(x_{1:s}) : s < t\}.$$

Definition 3.1 forms the basis for subsequent notions of generation with replay.

### 3.2.1. UNIFORM GENERATION WITH REPLAY.

We begin with the most restrictive setting, where the generator must succeed after seeing a fixed number of samples $d^\star$, independent of the target $h$.

**Definition 3.2** (Uniform generatability with replay)**.** A class $\mathcal{H}$ is *uniformly generatable with replay* if there exist a generator $\mathcal{G}$ and $d^\star \in \mathbb{N}$ such that for every $h \in \mathcal{H}$ and every replay sequence $(x_t)_{t \geq 1}$ for $h$ and $\mathcal{G}$, if there exists $t$ with $|\{x_1, \ldots, x_t\}| = d^\star$, then $\mathcal{G}(x_{1:s}) \in$ supp $(h) \setminus \{x_1, \ldots, x_s\}$ for all $s \geq t$.

Given a generator $\mathcal{G}$, we define its *uniform generation with replay sample complexity $d^\star_{\mathcal{G}}$* as the smallest such $d^\star$, or $\infty$ if no such value exists.

> **Contribution 1.** In Theorem 4.1, we prove that $\mathcal{H}$ is uniformly generatable with replay if and only if it is uniformly generatable in the standard setting. Moreover, the sample complexity $d^\star$ remains unchanged.

Our proof proceeds by a black-box reduction from a uniform generator in the standard setting to one in the replay setting, showing how to make uniform generators sufficiently robust to absorb the noise introduced by replay.

### 3.2.2. NON-UNIFORM GENERATION WITH REPLAY

In this notion of generatability, the sample complexity $d^\star_h$ is allowed to depend on the target $h$ but not on the specific sequence of examples $(x_t)_{t \geq 1}$.

**Definition 3.3** (Non-uniform generatability with replay)**.** A class $\mathcal{H}$ is *non-uniformly generatable with replay* if there exists a generator $\mathcal{G}$ such that for every $h \in \mathcal{H}$ there exists $d^\star_h \in \mathbb{N}$ such that, for any sequence with replay $(x_t)_{t \geq 1}$ for $h$ and $\mathcal{G}$, if there exists $t^\star_h \in \mathbb{N}$ such that $\left|\left\{x_1, \ldots, x_{t^\star_h}\right\}\right| = d^\star_h$, then $\mathcal{G}(x_{1:s}) \in$ supp $(h) \setminus \{x_1, \ldots, x_s\}$ for all $s \geq t^\star_h$.

**Contribution 2.** In Theorem 5.1, we construct a *countable* hypothesis class that is non-uniformly generatable in the standard setting but is *not* non-uniformly generatable with replay.

In the standard setting, every countable class is non-uniformly generatable (Raman et al., 2025; Charikar & Pabbaraju, 2025). Thus, our result creates a strong separation in the non-uniform generation setting.

### 3.2.3. GENERATION IN THE LIMIT WITH REPLAY

This setting requires success only on example streams that eventually enumerate the entire support of the target hypothesis (possibly interleaved with replay samples), with no pre-computed bound on the sample complexity.

**Definition 3.4** (Generatability in the limit with replay). An infinite replay sequence $(x_t)_{t \geq 1}$ is an *enumeration with replay* if it eventually reveals every $x \in \mathrm{supp}(h)$.[1] A class $\mathcal{H}$ is *generatable in the limit with replay* if there exists a generator $\mathcal{G}$ such that, for every $h \in \mathcal{H}$ and every enumeration with replay $(x_t)_{t \geq 1}$, there exists $t^\star \in \mathbb{N}$ such that $\mathcal{G}(x_{1:s}) \in \mathrm{supp}(h) \setminus \{x_1, \ldots, x_s\}$ for all $s \geq t^\star$.

Crucially, to meet the requirement of enumerating the full support of $h^\star$, the adversary can reveal an instance *after* it has been output by the generator. This increases the hardness of generation, as the generator must carefully select its outputs, knowing that replayed instances cannot be trusted.

**Contribution 3.1.** In Theorem 6.2, we prove that there exists an (uncountable) class $\mathcal{H}$ that is generatable in the limit without replay but is *not* generatable in the limit with replay.

This separation shows that the replay model can fundamentally limit the power of generation over general hypothesis classes. This naturally raises the question of whether a similar separation holds for *countable* classes. For this specific case, we provide a positive result.

**Contribution 3.2.** In Theorem 6.1, we provide an algorithm that generates in the limit under replay any countable class using only membership queries.

By *membership queries* we mean oracle access to the predicate "$x \in \mathrm{supp}(h)$" for any $h \in \mathcal{H}$ and any $x \in \mathcal{X}$.[2] Kleinberg & Mullainathan (2024) showed that, in the standard setting, every countable class is generatable in the limit using only membership queries. Thus, our result shows that, for countable classes, generation in the limit remains equally possible under replay using the same access model.

### 3.2.4. PROPER GENERATION WITH AND WITHOUT REPLAY

Finally, we study *proper* generation, where at each round the generator must output a hypothesis $\hat{h}_t \in \mathcal{H}$, rather than an element $o_t \in \mathcal{X}$, and the success criterion requires

$$\mathrm{supp}\left(\hat{h}_t\right) \subseteq \mathrm{supp}(h^\star)$$

for all sufficiently large $t$. We refer to the setting where the generator outputs elements of $\mathcal{X}$ simply as *generation*, and occasionally as *improper* generation when a contrast with the proper setting is needed.[3]

The replay adversary may now reveal *any* element from the support of any previously output hypothesis. This formalizes unconstrained downstream reuse of deployed generative models, where each $\hat{h}_t$ represents a specific version of a model previously deployed and accessible to downstream users for content generation.

**Definition 3.5** (Proper generatability in the limit with replay). A class $\mathcal{H}$ is *properly generatable in the limit with replay* if there exists a proper generator $\mathcal{G}$ such that, for every $h \in \mathcal{H}$ and every sequence $(x_t)_{t \geq 1}$ satisfying

1. $x_t \in \mathrm{supp}(h)$ or $x_t \in \mathrm{supp}\left(\hat{h}_s\right)$ for some $s < t$, and
2. $(x_t)_{t \geq 1}$ enumerates every element of $\mathrm{supp}(h)$,

there exists $t^\star$ such that $\mathrm{supp}\left(\hat{h}_t\right) \subseteq \mathrm{supp}(h)$ for all $t \geq t^\star$.

**Contribution 4.1.** In Theorem 7.2, we show that there exists a *finite* class $\mathcal{H}$ that is properly generatable in the limit in the standard setting, but *not* in the replay setting.

Table 1 summarizes the results described so far, highlighting whether replay affects the guarantees of the standard setting for each notion of generatability.

We also show that, even without replay, proper generation in the limit can be strictly harder than improper generation in a computational sense.

**Contribution 4.2.** In Theorem 7.1, we show that proper generation in the limit may require stronger computational primitives than membership queries alone.

---

[1] That is, for every $x \in \mathrm{supp}(h)$ there exists a finite $t \in \mathbb{N}$ such that $x_t = x$.

[2] Equivalently, we assume that $h(x)$ is computable for all $h \in \mathcal{H}$ and $x \in \mathcal{X}$. We note that this access model is, in a sense, minimal, as any reasonable generator should at least be able to evaluate whether any given $h \in \mathcal{H}$ is consistent with the example stream.

[3] In the literature, *improper* and *proper* generation are also referred to as *element-based* and *index-based* generation, respectively (Kleinberg & Wei, 2025). The term *index-based*, however, presumes that $\mathcal{H}$ is countable and thus admits an indexing.

Kleinberg & Mullainathan (2024) showed that, in the standard setting, all countable classes are properly generatable in the limit using membership queries and subset queries.[4] Theorem 7.1 establishes a computational lower bound for the algorithm of Kleinberg & Mullainathan (2024), showing that access to some additional computational primitive (such as subset queries) is necessary in general. This result is of independent interest beyond generation with replay.

## 4. Uniform Generation with Replay

We begin with the simplest result. In the standard setting, a uniformly generatable class $\mathcal{H}$ has the nice property that its sample complexity $d^\star$ is known, at least information-theoretically (Raman et al., 2025). That is, after observing any $d^\star$ distinct examples from any given $h \in \mathcal{H}$, a uniform generator $\mathcal{G}$ for $\mathcal{H}$ is guaranteed to output unseen elements of $\mathrm{supp}\,(h)$. A naive strategy to convert $\mathcal{G}$ into a generator $\tilde{\mathcal{G}}$ that generates $\mathcal{H}$ uniformly with replay is to ignore all examples matching a previous output and apply $\mathcal{G}$ on the remaining examples. However, if $\tilde{\mathcal{G}}$ were to output arbitrary elements $o_t \in \mathcal{X}$, then the sequence

$$x_1, \quad o_1, \quad o_2, \quad o_3, \quad \ldots$$

could, in principle, form a valid replay sequence with potentially unbounded cardinality. In this case, $\tilde{\mathcal{G}}$ would not gather any additional information on $h^\star$ beyond $x_1 \in \mathrm{supp}\,(h^\star)$, and thus would not automatically inherit $\mathcal{G}$'s guarantees.

To address this challenge, we introduce a preliminary "burn-in" phase during which we restrict $\tilde{\mathcal{G}}$'s outputs, before eventually copying $\mathcal{G}$. Algorithm 1 illustrates how to construct such a generator $\tilde{\mathcal{G}}$ achieving uniform generation under replay in the most sample-efficient way possible. This result is stated formally in the following theorem.

**Theorem 4.1** (Equivalence of uniform generation with and without replay). *A binary hypothesis class $\mathcal{H} \subseteq \{0,1\}^{\mathcal{X}}$ satisfying the UUS property is uniformly generatable with replay if and only if it is uniformly generatable. In particular, any generator $\mathcal{G}$ that generates $\mathcal{H}$ uniformly can be converted into a generator $\tilde{\mathcal{G}}$ that generates $\mathcal{H}$ uniformly with replay, without increasing the sample complexity.*

*Proof.* Clearly, if a generator generates $\mathcal{H}$ uniformly with replay then it also generates $\mathcal{H}$ uniformly, since all valid sequences in the standard setting are also valid sequences with replay. To show the other implication, suppose $\mathcal{G}$ generates $\mathcal{H}$ uniformly after seeing $d^\star$ examples. Algorithm 1 shows how to construct a generator $\tilde{\mathcal{G}}$ from $\mathcal{G}$ that generates $\mathcal{H}$ uniformly with replay. Let $(x_t)_{t \geq 1}$ be a sequence

---

[4]By *subset queries* we mean queries of the kind "$\mathrm{supp}\,(h_i) \subseteq \mathrm{supp}\,(h_j)$?" for any $h_i, h_j \in \mathcal{H}$.

---

**Algorithm 1** Uniform-to-uniform-with-replay conversion

**Require:** $\mathcal{G}$ uniform generator with sample complexity $d^\star$
1: **for** $t = 1, 2, \ldots$ **do**
2:     Receive new example $x_t$
3:     **if** $|\{x_1, \ldots, x_t\}| \geq d^\star$ **then**
4:         Output $\mathcal{G}(x_{1:t})$
5:     **else**
6:         Output $x_1$
7:     **end if**
8: **end for**

---

with replay for $h \in \mathcal{H}$ and $\tilde{\mathcal{G}}$. The generator $\tilde{\mathcal{G}}$ repeatedly outputs the first example $x_1$ until the following condition is satisfied: $|\{x_1, \ldots, x_t\}| \geq d^\star$. From that moment onward, $\tilde{\mathcal{G}}$ copies $\mathcal{G}$'s outputs. Now, let $t^\star$ be the first time such that $|\{x_1, \ldots, x_{t^\star}\}| \geq d^\star$. We necessarily have that $\{x_1, \ldots, x_{t^\star}\} \subset \mathrm{supp}\,(h)$, since $x_1$ is guaranteed to belong to $\mathrm{supp}\,(h)$ and $\tilde{\mathcal{G}}$ has only ever output $x_1$. Therefore, since $\mathcal{G}$ uniformly generates $\mathcal{H}$ with sample complexity $d^\star$, we have that $\mathcal{G}(x_{1:s}) \in \mathrm{supp}\,(h) \setminus \{x_1, \ldots, x_s\}$ for all $s \geq t^\star$. It follows that $\tilde{\mathcal{G}}$ achieves uniform generation with replay with the same sample complexity $d^\star$. $\square$

We now turn to a setting where the previous approach fails and introducing replay yields a strict separation from the standard setting.

## 5. Non-Uniform Generation with Replay

In contrast to the uniform notion of generation, the sample complexity in the non-uniform case depends on the particular, *unknown* target hypothesis (see Definition A.4 in Appendix A). As a result, a generator cannot commit in advance to observing a fixed number of distinct examples before producing new outputs, as in Section 4. This precludes a direct adaptation of the reduction-based constructions from uniform generators used in, e.g., Raman et al. (2025).

In the standard setting, all countable classes are non-uniformly generatable (Raman et al., 2025; Charikar & Pabbaraju, 2025). In contrast, Theorem 5.1 shows that this guarantee fails in the replay setting: countability alone no longer suffices. Nonetheless, every finite hypothesis class remains non-uniformly generatable as an immediate corollary of Theorem 4.1. Together, these results account for the row on non-uniform generation in Table 1.

**Theorem 5.1** (Hardness of non-uniform generation with replay). *There exists a* countable *binary hypothesis class $\mathcal{H} \subseteq \{0,1\}^{\mathcal{X}}$ satisfying the UUS property that is* not *non-uniformly generatable with replay.*

*Proof.* Let $\mathcal{X} = \mathbb{Z}$. For each $n \in \mathbb{N}$ define the hypotheses

$h_n$ and $h_\infty$ by

$$\text{supp}\,(h_n) = \{1, \ldots, n\} \cup \mathbb{Z}_{<0}, \quad \text{supp}\,(h_\infty) = \mathbb{N}.$$

Let $\mathcal{H} := \{h_\infty\} \cup \{h_n : n \in \mathbb{N}\}$. Assume for contradiction that there exists a generator $\mathcal{G}$ that non-uniformly generates $\mathcal{H}$ with replay. Let $d := d^\star_{h_\infty}$ denote the (non-uniform) sample complexity associated with $h_\infty$.

We define an adversarial sequence $(x_t)_{t \geq 1}$ online. For $t = 1, \ldots, d$, set $x_t := t$. For each $t \geq d$, set $x_{t+1} := o_t$, i.e., from time $d$ onward the adversary always replays the most recent generator's output $o_t := \mathcal{G}\,(x_{1:t})$. By construction, $(x_t)_{t \geq 1}$ is a valid replay sequence for $h_\infty$ and $\mathcal{G}$: the first $d$ points lie in $\text{supp}\,(h_\infty)$ and every subsequent point is a replay.

Since $|\{x_1, \ldots, x_d\}| = d = d^\star_{h_\infty}$, generatability in the non-uniform setting implies that for all $t \geq d$,

$$o_t \in \text{supp}\,(h_\infty) \setminus \{x_1, \ldots, x_t\} = \mathbb{N} \setminus \{x_1, \ldots, x_t\}.$$

In particular, since $x_{t+1} = o_t$, it follows that the generator outputs *fresh* natural numbers from time $d$ onward. Thus, the set of distinct points in $(x_t)_{t \geq 1}$ is unbounded.

Next, observe that the same sequence $(x_t)_{t \geq 1}$ is also a valid sequence with replay for $h_d$ and $\mathcal{G}$: we have $1, \ldots, d \in \text{supp}\,(h_d)$, and for all later times the adversary supplies replays (which are allowed to lie outside the support of $h_d$). Because the sequence $(x_t)_{t \geq 1}$ contains infinitely many distinct points, there exists a finite $T \in \mathbb{N}$ such that $|\{x_1, \ldots, x_T\}| \geq d^\star_{h_d}$. Applying the non-uniform guarantee to the target $h_d$ therefore yields that, for all $t \geq T$,

$$o_t \in \text{supp}\,(h_d) \setminus \{x_1, \ldots, x_t\}.$$

Combining this with $o_t \in \text{supp}\,(h_\infty) = \mathbb{N}$ for all $t \geq d$, we obtain that for all $t \geq \max\{d, T\}$,

$$o_t \in \text{supp}\,(h_\infty) \cap \text{supp}\,(h_d) = \{1, \ldots, d\}.$$

Thus, for all sufficiently large $t$, the output $o_t$ must lie in the finite set $\{1, \ldots, d\}$ while also being fresh relative to $\{x_1, \ldots, x_t\}$. This is impossible: after at most $d$ such fresh outputs, every element of $\{1, \ldots, d\}$ has already appeared in the input sequence. The resulting contradiction shows that no such generator $\mathcal{G}$ can exist. □

# 6. Generation in the Limit with Replay

We first construct an algorithm that matches the computational guarantees of the standard setting for all countable classes under replay (Theorem 6.1). Then, in Theorem 6.2, we present a hard (necessarily uncountable) hypothesis class demonstrating a separation between generation in the limit with and without replay. See also the corresponding row of Table 1.

## 6.1. A Computable Algorithm to Generate in the Limit Any Countable Class under Replay

Kleinberg & Mullainathan (2024) show that all countable hypothesis classes are generatable in the limit *without* replay and give a universal membership-query-only generator. The next theorem shows that replay does not change this picture.

**Theorem 6.1.** *There exists a generator that, given any countable binary hypothesis class $\mathcal{H} = \{h_1, h_2, \ldots\}$ over a countable domain $\mathcal{X}$ satisfying the UUS property, generates in the limit with replay every target $h^\star \in \mathcal{H}$ using only membership queries.*

*Proof sketch of Theorem 6.1.* We defer the full proof of Theorem 6.1 and pseudocode to Appendix B; here we provide only a proof sketch. Consider WP (*Witness Protection*; Algorithm 2 in Appendix B), a replay-robust variant of the algorithm of Kleinberg & Mullainathan (2024). Since the domain $\mathcal{X}$ is countable, we may assume $\mathcal{X} = \mathbb{N}$. Fix a target hypothesis $h^\star = h_z$. The algorithm maintains a growing prefix length $m$ over the domain and compares hypotheses only on the finite prefix $\{1, \ldots, m\}$, so that all containment tests used by the algorithm are computable with membership queries alone. Recall that, in the replay model, an observed example is reliable only if it cannot be explained as a replay of a previous output. Thus, WP stores previous outputs inside the set $O_t$ and maintains the set $S_t$ of *sure* examples, which necessarily satisfies $S_t \subseteq \text{supp}\,(h^\star)$. The algorithm follows the critical-hypothesis strategy of Kleinberg & Mullainathan (2024), but with the consistency and containment tests modified for replay. A hypothesis is tested for consistency only against the sure set $S_t$. Among the hypotheses consistent with $S_t$, a hypothesis $h_i$ is $(t, m)$-*critical with replay* (Definition B.1) when, on the current finite prefix $m$, it is no larger than any earlier consistent hypothesis $h_j$, $j < i$, after discounting the elements $O_{t-1}$ that the generator has already output.

The algorithm selects the largest index $n^{(t,m)} \leq t$ of a $(t, m)$-critical hypothesis and attempts to output an element of $\text{supp}\,(h_{n^{(t,m)}}) \cap \{1, \ldots, m\}$ that is outside $S_t \cup O_{t-1}$ and also outside a set $W^{(t,m)}$ of *witnesses* that the algorithm never outputs. If no such element exists on the current prefix, WP increases $m$, recomputes criticality and witnesses on the larger prefix, and repeats. The witness set $W^{(t,m)}$ is constructed as follows. For each pair $j < i$ of currently consistent hypotheses, WP searches the current prefix for an element witnessing that $h_i$ contains something not contained in $h_j$, modulo previous outputs. Whenever such a witness is found, the algorithm protects it by never outputting it. Consequently, if that witness later appears in the stream, it cannot be a replay and therefore enters $S_t$, permanently eliminating $h_j$. Note that the per-round search of an admissible output is guaranteed to terminate because the selected hypothesis has infinite support while only finitely

many elements are forbidden on each round (Lemma B.3).

Now consider the true hypothesis $h_z$. For each earlier hypothesis $h_i$ with $i < z$ that is consistent with the sure set, either $\mathrm{supp}\,(h_z)$ is already contained in $\mathrm{supp}\,(h_i)$ up to previously output elements, or there is a witness in $\mathrm{supp}\,(h_z) \setminus (\mathrm{supp}\,(h_i) \cup O_{z-1})$. In the latter case, WP eventually protects such a witness. Since the input stream is an enumeration with replay of $\mathrm{supp}\,(h_z)$, the witness must eventually be presented; because it was protected, it enters the sure set $S_t$, so $h_i$ is ruled out. There are only finitely many indices $i < z$, so after finitely many eliminations every remaining earlier consistent hypothesis contains $h_z$ on every finite prefix, up to previous outputs. Equivalently, $h_z$ is eventually critical in the replay sense (Lemma B.2). From that time onward, the hypothesis selected by WP is either $h_z$ itself or a later critical hypothesis whose finite-prefix support is contained in $\mathrm{supp}\,(h_z)$ modulo previous outputs. Moreover, since WP outputs outside the sure set, outside the previous outputs, and outside the protected witnesses, the output is in $\mathrm{supp}\,(h_z)$ and has not appeared in the input stream. Therefore, WP generates $h_z$ in the limit with replay using only membership queries. Since $h_z$ was arbitrary, the result follows. □

### 6.2. Separation Between Generation in the Limit with and without Replay

While the previous result shows that replay does not impose additional hardness on the generatability in the limit of countable hypothesis classes, it leaves open whether replay can ever make generation strictly harder. The following theorem answers this in the affirmative: there are (uncountable) classes that are generatable in the limit in the standard sense but not generatable in the limit when replay is allowed.

**Theorem 6.2.** *There exists a hypothesis class $\mathcal{H}$ that is generatable in the limit but is not generatable in the limit with replay.*

*Proof sketch of Theorem 6.2.* We defer the proof to Appendix C and only present the high-level idea here. Let $\mathcal{X} = \mathbb{Z} \cup \{*^n : n \in \mathbb{N}\}$. For each $b \in \mathbb{N}_0$, let $\mathcal{H}^b$ be a class introduced by Bai et al. (2026) defined as the union of two subfamilies: one whose hypotheses include $b$ and eventually all large integers beyond some cutoff, and one whose hypotheses omit $b$ but contain all integers smaller than $b$; in both subfamilies, all hypotheses additionally contain an arbitrary subset of $\mathbb{Z}$, making the class uncountable. The key property of $\mathcal{H}^b$ is that it is generatable in the limit in the standard setting but not if the enumeration $(x_t)_{t \geq 1}$ is allowed to omit the instance $b$. We build the hard class $\mathcal{H}$ by taking the union of $\mathcal{H}^b$ over $b \in \mathbb{N}_0$. Moreover, we *tag* each component $\mathcal{H}^b$ using special *marker* instances: every hypothesis in $\mathcal{H}^b$ additionally contains the finite marker

set $\{*^k : 1 \leq k \leq b\}$. The final class $\mathcal{H}$ also includes the all-marker hypothesis, whose support is $\{*^k : k \in \mathbb{N}\}$.

These markers ensure that $\mathcal{H}$ is generatable in the limit without replay (Lemma C.1). Indeed, from any enumeration $(x_t)_{t \geq 1}$ of any target $h^\star$ belonging to some $\mathcal{H}^b$, the largest marker observed so far is nondecreasing and eventually stabilizes at $*^b$. The generator can then restrict to $\mathcal{H}^b$ and run the corresponding in-the-limit generator for $\mathcal{H}^b$ on the integer part of the stream: observing $b$ certifies that the target is in the first subfamily, allowing the generator to switch to producing fresh large integers; without observing $b$, the safe outputs are instead integers below $b$. If the target is the all-marker hypothesis, the generator instead keeps outputting fresh markers.

In the replay setting, however, the adversary can prevent $\mathcal{G}$ from ever certifying which $\mathcal{H}^b$ the target $h^\star$ belongs to. Since $\mathcal{H}$ contains the all-marker hypothesis, the adversary can force any generator $\mathcal{G}$ that generates $\mathcal{H}$ in the limit under replay to eventually output a fresh marker $*^z$. Because $*^z$ was first produced by $\mathcal{G}$, its later appearance in the enumeration does not certify that $*^z \in \mathrm{supp}\,(h^\star)$. Thus, replay keeps the enumeration $(x_t)_{t \geq 1}$ simultaneously consistent with both some hypothesis in $\mathcal{H}^{z-1}$ and some hypothesis in $\mathcal{H}^z$. The adversary exploits this ambiguity to force $\mathcal{G}$ to commit infinitely many mistakes. Lemma C.2 formalizes this indistinguishability via a diagonalization argument. □

## 7. Proper Generation in the Limit

We now shift our focus from *improper* to *proper* generation, where the generator outputs a hypothesis $\hat{h}_t$ at each round. We focus exclusively on the *in-the-limit* notion of proper generatability for two reasons. First, prior work on proper generatability has primarily addressed the in-the-limit setting: Kleinberg & Mullainathan (2024) established that all countable classes are properly generatable in the limit using a generator relying on membership and subset queries (see Theorem A.15). In Section 7.1, we strengthen this line of work by providing a computational lower bound showing that membership queries alone are insufficient for proper generation in the limit in the standard setting. Second, as we show in Section 7.2, the notion of proper generatability with replay is so strong that a separation from the standard setting arises even in the (easy) setting of generatability in the limit of finite classes.

### 7.1. An Impossibility Result for Proper Generation in the Limit Using Only Membership Queries

Kleinberg & Mullainathan (2024) give a universal membership-query-only algorithm that improperly generates in the limit any countable hypothesis class. A similar algorithm also achieves *proper* generation in the limit for

any countable class, but requires additional access to subset queries. The following result shows that access to additional queries besides membership queries is indeed *necessary* for proper generation.

**Theorem 7.1.** *There cannot exist a (deterministic) generator $\mathcal{G}$ that only makes membership queries and properly generates in the limit all countable hypothesis classes.*

*Proof sketch of Theorem 7.1.* We provide only a proof sketch here and defer the full proof to Appendix D. Fix a deterministic proper generator $\mathcal{G}$ that only makes membership queries. We construct a hard class $\mathcal{H}$ by simulating $\mathcal{G}$'s interaction with an adversarial enumeration. Our proof uses the same high-level simulation strategy as the computational lower bound of Charikar & Pabbaraju (2025), with the additional complication that our construction maintains a *countably infinite* hypothesis class rather than two hypotheses. Algorithm 3 (in Appendix D) defines $\mathcal{H} = \{h_1, h_2, \ldots\}$ via a function $F : \mathbb{N} \times \mathbb{N} \to \{0, 1\}$, where

$$F(i, j) = 1 \iff j \in \mathrm{supp}\,(h_i).$$

The algorithm maintains counters $I, J$ delimiting the finite rectangle of hypothesis-instance pairs $(i, j)$ queried so far by $\mathcal{G}$. Outside this rectangle, memberships can be fixed adversarially. To ensure that all elements of the target hypothesis are enumerated at a finite time, the algorithm also maintains a queue $Q$ of elements that need to be shown and at each round reveals $x_t = \min Q$. Lemma D.1 shows that the resulting $\mathcal{H}$ is a valid countable class satisfying the UUS property.

The construction has two modes, *diagonalization* and *overgeneralization*, and it switches modes automatically through the enumeration, driven by the hypotheses output by $\mathcal{G}$. It maintains a distinguished hypothesis $h_1$ and a trap pair $(i', j')$ such that $j' \notin \mathrm{supp}\,(h_{i'})$ but $j' \in \mathrm{supp}\,(h_i)$ for all $i \neq i'$. Crucially, the current trap instance $j'$ is added to $Q$ only when $\mathcal{G}$ outputs $\hat{h}_t \neq h_1$. We consider two cases:

- If $\mathcal{G}$ outputs $\hat{h}_t \neq h_1$ infinitely often, then each such round triggers a *diagonalization* step: the algorithm creates a fresh instance $d_t > J$ with $d_t \in \mathrm{supp}\,(\hat{h}_t)$ but $d_t \notin \mathrm{supp}\,(h_i)$ for all $h_i \neq \hat{h}_t$. Thus, in particular, $\mathrm{supp}\,(\hat{h}_t) \nsubseteq \mathrm{supp}\,(h_1)$ on infinitely many rounds. Meanwhile, the revealed sequence enumerates $\mathrm{supp}\,(h_1)$: instances assigned to all hypotheses are enqueued immediately, and each trap instance is enqueued the next time $\hat{h}_t \neq h_1$. Thus, taking $h^\star = h_1$, the generator makes infinitely many mistakes.

- Otherwise, $\mathcal{G}$ outputs $\hat{h}_t \neq h_1$ only finitely many times. That is, there exists a last deviation time $t'$ such that $\hat{h}_{t'} \neq h_1$ but $\hat{h}_t = h_1$ for all $t > t'$. Let

$(i', j')$ be the trap pair at the end of round $t'$. Since no further deviations occur, $j'$ is never added to $Q$. All other instances belonging to $\mathrm{supp}\,(h_{i'})$ are eventually enqueued, either immediately or during one of the finitely many earlier deviations. Thus, the revealed sequence enumerates $\mathrm{supp}\,(h_{i'})$. Taking $h^\star = h_{i'}$, $\mathcal{G}$ *overgeneralizes* forever since $\mathrm{supp}\,(h_1) \nsubseteq \mathrm{supp}\,(h_{i'})$.

In both cases, $\mathcal{G}$ fails to properly generate in the limit. $\square$

### 7.2. Proper Generation in the Limit with Replay

The following theorem shows that, in the proper setting, replay makes a class of just four hypotheses not generatable under even the weakest notion. This accounts for the last row of Table 1.

**Theorem 7.2** (Hardness of proper generation in the limit with replay)**.** *There exists a* finite *hypothesis class $\mathcal{H}$ that is not properly generatable in the limit with replay.*

*Proof.* For $i = 1, 2$, define

$$\mathrm{supp}\,(h_i^-) = \mathbb{Z}_{\leq 0} \cup \{i\}, \quad \mathrm{supp}\,(h_i^+) = \mathbb{Z}_{\geq 0} \cup \{-i\},$$

and let $\mathcal{H} = \{h_1^-, h_2^-, h_1^+, h_2^+\}$. Suppose, for the sake of contradiction, that there exists a proper generator $\mathcal{G}$ that properly generates $\mathcal{H}$ in the limit with replay. Let $x_1 = 0$ be the first example shown by the adversary. Note that $x_1$ belongs to the support of all hypotheses in $\mathcal{H}$. Therefore, $\mathcal{G}$ makes a completely arbitrary choice when choosing its first output $\hat{h}_1$. We give the argument for $\hat{h}_1 = h_1^-$; the other cases are handled analogously.

Consider the following extension of the adversarial sequence of examples: $x_2 = -1, x_3 = -2$, followed by all the positive integers. The resulting sequence $(x_t)_{t \geq 1}$ is a valid sequence with replay for $\mathcal{G}$ and both $h_1^+, h_2^+$:

$$x_t \in \mathrm{supp}\,(h_1^+) \cap \mathrm{supp}\,(h_2^+) \text{ for } t \neq 2, 3$$

and

$$x_2, x_3 \in \mathrm{supp}\,(\hat{h}_1).$$

Additionally, $(x_t)_{t \geq 1}$ contains an enumeration of the support of both $h_1^+$ and $h_2^+$ and, thus, is an enumeration with replay in the proper setting for $h_1^+$ and $h_2^+$ simultaneously. As $\mathcal{G}$ properly generates $\mathcal{H}$ in the limit with replay by assumption, there exist $t_1^\star, t_2^\star \in \mathbb{N}$ and a sequence of $\hat{h}_t \in \mathcal{H}$ such that:

$$\mathrm{supp}\,(\hat{h}_t) \subseteq \mathrm{supp}\,(h_1^+) \text{ for all } t \geq t_1^\star$$

and

$$\mathrm{supp}\,(\hat{h}_t) \subseteq \mathrm{supp}\,(h_2^+) \text{ for all } t \geq t_2^\star.$$

Therefore, if we let $t^\star = \max\{t_1^\star, t_2^\star\}$, it must be that, for all $t \geq t^\star$,

$$\mathrm{supp}\left(\hat{h}_t\right) \subseteq \mathrm{supp}\left(h_1^+\right) \cap \mathrm{supp}\left(h_2^+\right) = \mathbb{Z}_{\geq 0}.$$

However, there is no hypothesis $h \in \mathcal{H}$ such that $\mathrm{supp}\,(h) \subseteq \mathbb{Z}_{\geq 0}$; we have reached a contradiction. $\qquad\square$

## 8. Discussion and Open Questions

This work asks when replay makes generation harder. The answer turns out to depend on the specific notion of generation and on the complexity of the hypothesis class, with qualitatively different outcomes across settings. Nonetheless, our positive results are driven by a common set of intuitions, centered around two key ideas:

- *Data cleaning and watermarking.* Algorithm 2 treats potentially replayed instances as misleading and discards them. By considering a deterministic generator—as is standard in much of the language generation literature—we implicitly assume access to the information needed to identify such instances. From a practical standpoint, this motivates data provenance measures, watermarking, as well as the curation of clean training datasets (Kirchenbauer et al., 2023; 2024; Sadasivan et al., 2023; Mitchell et al., 2023; Dathathri et al., 2024; Tang et al., 2024; Wu et al., 2025). However, reliable and scalable filtering is nontrivial in practice. This raises the question of whether structural properties of the hypothesis class could be leveraged to avoid explicit filtering altogether.

- *Output filtering.* Our algorithms impose strict constraints on the generator's output: Algorithm 1 has a preliminary burn-in phase during which it only outputs a dummy element; Algorithm 2 avoids outputting a set of crucial elements dubbed "witnesses" to ensure that, once such instances are shown as examples, their trustworthiness is guaranteed and they need not be discarded. However, these constraints may be at odds with the requirement that LLM outputs remain diverse, a property often referred to as *breadth* in the language generation literature (Kleinberg & Mullainathan, 2024). Therefore, a natural next step is to examine how replay affects not only the feasibility of generation, as studied in this work, but also the ability to generate with breadth.

Taken together, our positive results provide a theoretical lens on why such strategies, involving both input and output filtering, can be effective at mitigating model collapse. The separations, on the other hand, identify worst-case regimes in which these strategies may be insufficient.

This interpretation should, however, be read within the scope of the modeling assumptions of the *language generation in the limit* framework, which abstracts away many aspects of real LLM training pipelines, including the next-token prediction loss function, gradient-based optimization, and transformer architectures. The framework's agnosticism to such implementation details is both a limitation and a strength. While it is harder to directly prescribe implementation-level feedback for practitioners, it allows us to identify fundamental possibility and impossibility boundaries that are not tied to any particular architecture or optimization procedure, and may therefore remain relevant if we move away from current generation paradigms.

Some important aspects of real data contamination are also absent from our replay model. We assume exact replay of past outputs, whereas synthetic content in practice may be paraphrased, edited, mixed with human-written data, or produced by different models. Moreover, we adopt a worst-case adversarial model, whereas real data collection is not typically adversarial in this strong sense. While this worst-case formulation enables a direct comparison with the results in the standard setting from the language generation literature, more realistic variants are worth studying, including replay combined with external contamination (e.g., arbitrary insertions or omissions) and stochastic replay. For instance, in the proper setting, one could consider a stochastic model where the adversary can replay only a randomly selected element from a previously output hypothesis, which may help circumvent the strong impossibility result established in Theorem 7.2.

In addition to the directions outlined above, several technical questions remain open. A natural next step is to characterize non-uniform generatability under replay, since such a characterization exists for non-uniform generation in the standard setting (Raman et al., 2025). Another direction is to study randomized generators, which are not covered by our deterministic framework (except after fixing the random seed). Finally, our results motivate a more systematic study of proper generation from both information-theoretic and computational perspectives, since proper generation more directly models the sequential deployment and updating of generative models.

## Acknowledgements

GR and AS acknowledge the Novo Nordisk Foundation for support via the Startup grant (NNF24OC0087820); AS additionally acknowledges support from VILLUM FONDEN via the Young Investigator program (VIL72069). The authors also thank Carolin Heinzler for helpful feedback, as well as the anonymous reviewers for their constructive comments.

## Impact Statement

This paper presents work whose goal is to advance the field of Machine Learning. There are many potential societal consequences of our work, none which we feel must be specifically highlighted here.

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

# A. Relevant Prior Work on Language Generation

Motivated by the recent success of large language models, the study of *language generation in the limit* builds on the classical theory of *language identification from positive data*, initiated by Gold (1967) and characterized by Angluin (1980). While that line of work showed that identifying an unknown language is possible only under restrictive conditions, Kleinberg & Mullainathan (2024) showed that the weaker task of eventually generating new valid strings is possible for every countable language class. This striking contrast has since sparked a growing line of follow-up work on the learning-theoretic foundations of generation (Raman et al., 2025; Charikar & Pabbaraju, 2025; Kalavasis et al., 2025; Hanneke et al., 2025). Here, we summarize the prior work most relevant to our paper by first introducing the main definitions and then recalling the key results for each notion of generatability. Table 3 provides a side-by-side comparison with our results.

## A.1. Definitions

Let $\mathcal{X}$ be any infinite countable domain representing the space of possible outputs, such as text, images, or molecules, and let $\mathcal{H} \subseteq \{0,1\}^{\mathcal{X}}$ be a binary hypothesis class. Each hypothesis $h \in \mathcal{H}$ can be identified with its support $\operatorname{supp}(h) := \{x \in \mathcal{X} : h(x) = 1\}$, which represents the set of valid outputs according to $h$. For instance, in the context of text generation, $\mathcal{X}$ can be the set of all possible sentences, and each $h \in \mathcal{H}$ can be thought of as a language, with $\operatorname{supp}(h)$ being the set of admissible sentences. The language generation game takes place over infinite rounds between an adversary and a generator. The generator is defined as follows.

**Definition A.1** (Generator). A *generator* is a map that takes as input any finite sequence[5] $x_{1:t} := (x_1, \ldots, x_t) \in \mathcal{X}^t$ of any length $t \in \mathbb{N}$ and outputs an element $o_t \in \mathcal{X}$.

We first give an informal description of the game, which will be made formal in the definitions that follow. At the beginning of the game, the adversary chooses a target hypothesis $h^\star \in \mathcal{H}$ and a sequence of examples $x_1, x_2, \ldots$ such that $\{x_1, x_2, \ldots\} \subseteq \operatorname{supp}(h^\star)$. At each round $t$, the generator receives the prefix $x_{1:t}$ and produces an output $o_t$. The goal of the generator is to eventually produce an output that is *valid*, meaning that it belongs to $\operatorname{supp}(h^\star)$, and *novel*, meaning that it is not among the examples $x_1, \ldots, x_t$ seen so far. For this to be possible, the target hypothesis $h^\star$ must have infinite support, and hence we will only consider classes $\mathcal{H}$ that satisfy the following property.

**Definition A.2** (Uniformly unbounded support, UUS). A binary hypothesis class $\mathcal{H} \subseteq \{0,1\}^{\mathcal{X}}$ satisfies the *uniformly unbounded support (UUS)* property if $|\operatorname{supp}(h)| = \infty$ for all $h \in \mathcal{H}$.

We can now formally introduce the three main notions of generatability: *uniform*, *non-uniform*, and *in the limit*. As illustrated by Table 2, these notions differ on the restrictions placed on the success time $t^\star$ after which the generator must produce valid and novel outputs. We begin with the strongest notion, uniform generatability, where $t^\star$ is allowed to depend only on the hypothesis class $\mathcal{H}$, and therefore must hold uniformly over all hypotheses and example sequences.

**Definition A.3** (Uniform generatability, Kleinberg & Mullainathan 2024; Raman et al. 2025). A binary hypothesis class $\mathcal{H} \subseteq \{0,1\}^{\mathcal{X}}$ satisfying the UUS property is *uniformly generatable* if there exist a generator $\mathcal{G}$ and $d^\star \in \mathbb{N}$ such that, for every $h \in \mathcal{H}$ and any sequence $x_1, x_2, \ldots$ with $\{x_1, x_2, \ldots\} \subseteq \operatorname{supp}(h)$, if there exists $t^\star \in \mathbb{N}$ with $|\{x_1, \ldots, x_{t^\star}\}| = d^\star$, then $\mathcal{G}(x_{1:s}) \in \operatorname{supp}(h) \setminus \{x_1, \ldots, x_s\}$ for all $s \geq t^\star$.

For a given generator $\mathcal{G}$, its *uniform generation sample complexity* $d_{\mathcal{G}}^\star$ is defined as the smallest such $d^\star$, or $\infty$ if no such value exists.

Non-uniform generatability relaxes the uniformity requirement by allowing $t^\star$ to depend on the target hypothesis, but not on the example sequence.

**Definition A.4** (Non-uniform generatability, Raman et al. 2025). A binary hypothesis class $\mathcal{H} \subseteq \{0,1\}^{\mathcal{X}}$ satisfying the UUS property is *non-uniformly generatable* if there exists a generator $\mathcal{G}$ such that, for every $h \in \mathcal{H}$ there exists $d_h^\star \in \mathbb{N}$ such that, for any sequence $x_1, x_2, \ldots$ with $\{x_1, x_2, \ldots\} \subseteq \operatorname{supp}(h)$, if there exists $t^\star \in \mathbb{N}$ with $|\{x_1, \ldots, x_{t^\star}\}| = d_h^\star$, then $\mathcal{G}(x_{1:s}) \in \operatorname{supp}(h) \setminus \{x_1, \ldots, x_s\}$ for all $s \geq t^\star$.

For a given generator $\mathcal{G}$ and for any hypothesis $h \in \mathcal{H}$, the *non-uniform generation sample complexity* $d_{\mathcal{G},h}^\star$ is defined as the smallest such $d_h^\star$, or $\infty$ if no such value exists.

---

[5]In the replay setting, the specific *order* of the inputs matters since the adversary can present previous (potentially erroneous) outputs of the generator. For instance, let $x_2 = \mathcal{G}(x_1)$; the example sequences $(x_1, x_2)$ and $(x_2, x_1)$ may in principle contain different information in the replay setting.

*Table 2.* What is the success time $t^\star$ allowed to depend on?

| Generation notion | Class $\mathcal{H}$ | Hypothesis $h^\star$ | Sequence $(x_t)_{t \geq 1}$ |
|---|---|---|---|
| Uniform | Yes | No | No |
| Non-uniform | Yes | Yes | No |
| In the limit | Yes | Yes | Yes |

Finally, generatability in the limit is the weakest notion, where $t^\star$ is allowed to depend on the target hypothesis and on the example sequence.

**Definition A.5** (Generatability in the limit, Kleinberg & Mullainathan 2024). A binary hypothesis class $\mathcal{H} \subseteq \{0,1\}^{\mathcal{X}}$ satisfying the UUS property is *generatable in the limit* if there exists a generator $\mathcal{G}$ such that, for every $h \in \mathcal{H}$ and any *enumeration*[6] $(x_t)_{t \geq 1}$ of supp $(h)$, there exists $t^\star \in \mathbb{N}$ such that $\mathcal{G}(x_{1:s}) \in$ supp $(h) \setminus \{x_1, \ldots, x_s\}$ for all $s \geq t^\star$.

For any class $\mathcal{H}$, uniformly generatable $\implies$ non-uniformly generatable $\implies$ generatable in the limit.

Next, we describe the *proper* setting of generation, where, borrowing from the PAC learning vocabulary (Shalev-Shwartz & Ben-David, 2014), the algorithm is required to output a hypothesis $\hat{h}_t \in \mathcal{H}$ at each round $t$. In the terminology of the language generation literature (Kleinberg & Wei, 2025; Mehrotra et al., 2026), *proper* generation corresponds to *index-based* generation (at least for countable classes), while *improper* generation corresponds to *element-based* generation, though in this work we will often simply refer to the latter as *generation*.

**Definition A.6** (Proper generator, Kleinberg & Wei 2025). A *proper generator* is a map that takes as input any finite sequence $x_{1:t} := (x_1, \ldots, x_t) \in \mathcal{X}^t$ of any length $t \in \mathbb{N}$ and outputs a hypothesis $\hat{h}_t \in \mathcal{H}$.

**Definition A.7** (Proper generatability in the limit, Kleinberg & Wei 2025). A binary hypothesis class $\mathcal{H} \subseteq \{0,1\}^{\mathcal{X}}$ satisfying the UUS property is *properly generatable in the limit* if there exists a proper generator $\mathcal{G}$ such that, for all $h \in \mathcal{H}$ and for any enumeration $(x_t)_{t \geq 1}$ of supp $(h)$, there exists $t^\star \in \mathbb{N}$ such that supp $\left( \hat{h}_s \right) \subseteq$ supp $(h)$ for all $s \geq t^\star$.

Clearly, for any class $\mathcal{H}$, proper generatability $\implies$ (improper) generatability.

### A.2. Results

We now recall the standard guarantees against which our replay results will be compared, as illustrated by Table 3. We begin by recalling a combinatorial dimension that, for generatability, plays a role analogous to the VC dimension in PAC learning (Vapnik & Chervonenkis, 1971) and the Littlestone dimension in online learning (Littlestone, 1988).

**Definition A.8** (Closure dimension, Raman et al. 2025). The *Closure dimension* of a binary hypothesis class $\mathcal{H} \subseteq \{0,1\}^{\mathcal{X}}$, denoted by $\mathcal{C}(\mathcal{H})$, is the largest $d \in \mathbb{N}$ for which there exist distinct $x_1, \ldots, x_d \in \mathcal{X}$ such that there exists $h \in \mathcal{H}$ with $\{x_1, \ldots, x_d\} \subseteq$ supp $(h)$ and

$$\left| \bigcap_{h \in \mathcal{H}(x_{1:d})} \text{supp}(h) \right| < \infty,$$

where $\mathcal{H}(x_{1:d}) := \{h \in \mathcal{H} : \{x_1, \ldots, x_d\} \subseteq \text{supp}(h)\}$ denotes the version space. If this holds for any $d \in \mathbb{N}$, we say that $\mathcal{C}(\mathcal{H}) = \infty$; if it fails for $d = 1$, we say that $\mathcal{C}(\mathcal{H}) = 0$.

**Theorem A.9** (Characterization of uniform generatability, Raman et al. 2025). *A binary hypothesis class $\mathcal{H} \subseteq \{0,1\}^{\mathcal{X}}$ satisfying the UUS property is uniformly generatable if and only if $\mathcal{C}(\mathcal{H}) < \infty$.*

This leads to the following corollary for finite classes, which was first proved by Kleinberg & Mullainathan (2024).

**Corollary A.10.** *All finite hypothesis classes are uniformly generatable.*

The Closure dimension can also be used to characterize non-uniformly generatable classes, in a way that is reminiscent of how the VC dimension relates to non-uniform PAC learnability.

---

[6]An infinite sequence $(x_t)_{t \geq 1}$ is an enumeration of supp $(h)$ if for every $x \in$ supp $(h)$ there exists $t \in \mathbb{N}$ such that $x_t = x$.

*Table 3.* Generatability with and without replay.

| Generation notion | Without replay | With replay |
|---|---|---|
| Uniform | Theorem A.9[†] | Theorem 4.1[*] |
| Non-uniform | Theorem A.11[†], Theorem A.13[†] | Theorem 5.1[*] |
| In the limit | Theorem A.14[†] | Theorem 6.1[*], Theorem 6.2[*] |
| Proper in the limit | Theorem A.15[†], Theorem 7.1[*] | Theorem 7.2[*] |

[†] Prior work.    [*] New in this paper.

**Theorem A.11** (Characterization of non-uniform generatability, Raman et al. 2025). *A binary hypothesis class $\mathcal{H} \subseteq \{0,1\}^{\mathcal{X}}$ satisfying the UUS property is non-uniformly generatable if and only if there exists a non-decreasing sequence of classes $\mathcal{H}_1 \subseteq \mathcal{H}_2, \ldots$ such that $\mathcal{H} = \bigcup_{i=1}^{\infty} \mathcal{H}_i$ and $\mathcal{C}(\mathcal{H}_i) < \infty$ for every $i \in \mathbb{N}$.*

This immediately yields the following corollary for countable classes.

**Corollary A.12.** *All countable hypothesis classes are non-uniformly generatable.*

However, although all countable classes are non-uniformly generatable, there is an insurmountable computational barrier.

**Theorem A.13** (Non-uniform generation requires more than just membership queries, Charikar & Pabbaraju 2025). *A (deterministic) algorithm that non-uniformly generates all countable classes cannot be implemented using membership queries alone.*

The situation changes under generatability in the limit, where the following positive result holds for countable classes.

**Theorem A.14** (All countable classes are generatable in the limit using only membership queries, Kleinberg & Mullainathan 2024). *There exists a generator that generates all countable classes in the limit using only membership queries.*

Finally, in the *proper* setting, augmenting the same algorithm with subset queries yields the following result.

**Theorem A.15** (All countable classes are properly generatable in the limit, Kleinberg & Mullainathan 2024). *There exists a proper generator that properly generates all countable classes in the limit using membership queries and subset queries.*

## B. Proof of Theorem 6.1

We first restate the main theorem we are proving in this section.

**Theorem 6.1.** *There exists a generator that, given any* countable *binary hypothesis class $\mathcal{H} = \{h_1, h_2, \ldots\}$ over a countable domain $\mathcal{X}$ satisfying the UUS property, generates in the limit with replay every target $h^{\star} \in \mathcal{H}$ using only membership queries.*

The proof is constructive. Building on the algorithm of Kleinberg & Mullainathan (2024), we propose WP (*Witness Protection*; Algorithm 2), a universal membership-query-only algorithm that generates in the limit with replay any countable hypothesis class $\mathcal{H}$. To prove this, we first need some additional notation. Since the domain $\mathcal{X}$ is countable, we may assume without loss of generality that $\mathcal{X} = \mathbb{N}$. The algorithm maintains a growing prefix length $m$ over the elements of the domain $\mathcal{X}$. For $h \in \mathcal{H}$ and $m \in \mathbb{N}$, write

$$\mathrm{supp}(h)[m] := \mathrm{supp}(h) \cap \{1, \ldots, m\};$$

restricting hypotheses to this prefix yields a surrogate for set inclusion that is computable with membership queries alone.[7] Fix a target hypothesis $h^{\star} \in \mathcal{H}$. In the replay model, each example $x_t$ is either an element of $\mathrm{supp}(h^{\star})$ or a replay of a past output. Let $O_t := \{o_1, \ldots, o_t\}$ with $O_0 = \emptyset$, and define the *sure* set

$$S_t := \{x_s : 1 \leq s \leq t, \ x_s \notin O_{s-1}\},$$

---

[7]That is, for any $i, j, m \in \mathbb{N}$ we can establish whether $\mathrm{supp}(h_i)[m] \subseteq \mathrm{supp}(h_j)[m]$ using only a finite number of membership queries

i.e., the examples that cannot be explained as replay and hence must lie in $\text{supp}\,(h^\star)$, so that $S_t \subseteq \text{supp}\,(h^\star)$. We relax the notion of criticality from Kleinberg & Mullainathan (2024) to ignore previously output elements. Fix an ordering of the hypotheses of the countable class $\mathcal{H}$, i.e., write $\mathcal{H} = \{h_1, h_2, \ldots\}$.

**Definition B.1** $((t, m)$-critical with replay). Fix $t, m \in \mathbb{N}$. We say that $h_n \in \mathcal{H}$ is $(t, m)$-*critical with replay* if

1. $S_t \subseteq \text{supp}\,(h_n)$; and

2. for every $i < n$ with $S_t \subseteq \text{supp}\,(h_i)$ we have $\text{supp}\,(h_n)\,[m] \subseteq \text{supp}\,(h_i)\,[m] \cup O_{t-1}$.

Condition 1 requires that $h_n$ is consistent with the *sure* examples, i.e., examples that must belong to $\text{supp}\,(h^\star)$. Condition 2 enforces that on the finite prefix $\{1, \ldots, m\}$ of the domain, any earlier hypothesis $h_i$ that is also consistent with $S_t$ must contain every element of $\text{supp}\,(h_n)\,[m]$ except possibly those that could be explained as replays. Both conditions can be checked using finitely many membership queries.

At each step $t$, the algorithm considers the active set of consistent hypotheses

$$V_t := \{i \leq t : S_t \subseteq \text{supp}\,(h_i)\}.$$

From $V_t$, it selects the $(t, m)$-critical hypothesis with the largest index $n^{(t,m)}$ and attempts to output an element from

$$\text{supp}\,(h_{n^{(t,m)}})\,[m] \setminus \left( S_t \cup O_{t-1} \cup W^{(t,m)} \right),$$

where $W^{(t,m)}$ is the active *witness* set; the algorithm increases $m$ until a suitable element is found and then outputs it. To construct $W^{(t,m)}$, for any prefix $m$, active candidate set $V_t$, and pair $i, j \in V_t$ with $j < i$, define the witness $w_{ij}^{(t,m)}$ as the minimal unobserved element distinguishing $h_i$ and $h_j$ within the prefix $\{1, \ldots, m\}$:

$$w_{ij}^{(t,m)} := \min \Delta_{ij}^{(t,m)} \quad \text{where} \quad \Delta_{ij}^{(t,m)} := \text{supp}\,(h_i)\,[m] \setminus (\text{supp}\,(h_j)\,[m] \cup O_{t-1}),$$

when the set $\Delta_{ij}^{(t,m)}$ is not empty; otherwise, $w_{ij}^{(t,m)} = \perp$. The active witness set $W^{(t,m)}$ is then the collection of all such witnesses:

$$W^{(t,m)} := \left\{ w_{ij}^{(t,m)} \mid i, j \in V_t,\, j < i \right\} \setminus \{\perp\}.$$

Since the algorithm never outputs an active witness $w_{ij}^{(t,m)}$, if $w_{ij}^{(t,m)}$ appears in the example stream, it cannot be a replay and hence joins the sure set $S_t$, permanently ruling out $h_j$ from $V_t$.

Let $z$ be the first index with $h^\star = h_z$ in the given enumeration of $\mathcal{H}$. We prove the correctness of Algorithm 2 via three lemmas: (i) Lemma B.2 shows that $h_z$ eventually becomes $(t, m)$-critical with replay and stays so; (ii) Lemma B.3 shows that each round of Algorithm 2 terminates and the inner repeat-until loop finds an output in finite time; and finally (iii) Lemma B.4 shows that there exists a finite stabilization time $t^\star$ such that, for all steps after that, every output is fresh and valid for $h_z$.

**Lemma B.2** (Eventual criticality). *There exists $t^\star < \infty$ such that for all $t \geq t^\star$ and all $m \in \mathbb{N}$, the hypothesis $h_z$ is $(t, m)$-critical with replay.*

*Proof.* Consider step $t = z$ of Algorithm 2. Fix $j < z$ with $S_z \subseteq \text{supp}\,(h_j)$ and define

$$\Delta_{zj}^{(z,\infty)} := \text{supp}\,(h_z) \setminus (\text{supp}\,(h_j) \cup O_{z-1}).$$

If $\Delta_{zj}^{(z,\infty)} = \emptyset$, then $h_z$ already satisfies $\text{supp}\,(h_z) \subseteq \text{supp}\,(h_j) \cup O_{z-1}$. Otherwise, let $w_j := \min \Delta_{zj}^{(z,\infty)}$ (which is well-defined under the identification $\mathcal{X} \simeq \mathbb{N}$). Let

$$B := \left\{ j < z : S_z \subseteq \text{supp}\,(h_j) \text{ and } \Delta_{zj}^{(z,\infty)} \neq \emptyset \right\}.$$

Note that $|B| \leq z - 1 < \infty$. Also, recall that, since $h_z$ is the true hypothesis, $z \in V_t$ for all $t$.

---

**Algorithm 2** WITNESS PROTECTION (WP)

---

**Require:** $\mathcal{H} = \{h_1, h_2, \ldots\}$ over $\mathcal{X} = \{1, 2, \ldots\}$
1: $S_0 \leftarrow \emptyset; O_0 \leftarrow \emptyset; m \leftarrow 0$
2: **for** $t = 1, 2, \ldots$ **do**
3:     Receive a new example $x_t$
4:     **if** $x_t \notin O_{t-1}$ **then**
5:         $S_t \leftarrow S_{t-1} \cup \{x_t\}$
6:     **else**
7:         $S_t \leftarrow S_{t-1}$
8:     **end if**
9:     $V_t \leftarrow \{i \le t \mid S_t \subseteq \operatorname{supp}(h_i)\}$
10:     $o_t \leftarrow \bot$
11:     **if** $V_t = \emptyset$ **then**
12:         Choose $o_t \in S_t$ arbitrarily
13:     **else**
14:         $m \leftarrow \max\{m, x_t\}$
15:         **repeat**
16:             $m \leftarrow m + 1$
17:             $W^{(t,m)} \leftarrow \emptyset$
18:             **for** $i, j \in V_t$ with $j < i$ **do**
19:                 $\Delta_{ij}^{(t,m)} \leftarrow \operatorname{supp}(h_i)[m] \setminus (\operatorname{supp}(h_j)[m] \cup O_{t-1})$
20:                 **if** $\Delta_{ij}^{(t,m)} \neq \emptyset$ **then**
21:                     $w_{ij}^{(t,m)} \leftarrow \min \Delta_{ij}^{(t,m)}$
22:                     $W^{(t,m)} \leftarrow W^{(t,m)} \cup \left\{w_{ij}^{(t,m)}\right\}$
23:                 **end if**
24:             **end for**
25:             $n^{(t,m)} \leftarrow \max\{i \le t \mid h_i \text{ is } (t, m)\text{-critical with replay}\}$
26:             **for** $x \in \operatorname{supp}(h_{n^{(t,m)}})[m]$ **do**
27:                 **if** $x \notin S_t \cup O_{t-1} \cup W^{(t,m)}$ **then**
28:                     $o_t \leftarrow x;$ **break**
29:                 **end if**
30:             **end for**
31:         **until** $o_t \neq \bot$
32:     **end if**
33:     Output $o_t;$ $O_t \leftarrow O_{t-1} \cup \{o_t\}$
34: **end for**

---

We claim that, for every $j \in B$, $w_j \notin O_t$ as long as $j \in V_t$. We prove this by induction. For the base case $t = z - 1$, if $j \in B$, then by definition $w_j \notin O_{z-1}$. Now, consider any $t \ge z$. By the induction hypothesis, $w_j \notin O_{t-1}$. Thus, when $m \ge w_j$, WP sets

$$w_{zj}^{(t,m)} = \min\{x \le m : x \in \operatorname{supp}(h_z) \setminus (\operatorname{supp}(h_j) \cup O_{t-1})\} = w_j.$$

Since the output-selection rule forbids outputting any element of $W^{(t,m)}$, step $t$ does not output $w_j$, i.e., $w_j \notin O_t$. Otherwise, if $m < w_j$, then every output considered by the algorithm lies in $\{1, \ldots, m\}$, and hence cannot equal $w_j$. Thus, $w_j \notin O_t$ for any step $t$ where $j \in V_t$.

Since $w_j \in \operatorname{supp}(h_z)$, any enumeration with replay for $h_z$ and WP must eventually present $w_j$ at a finite time $t_j$ as some example $x_{t_j}$. There are two possibilities: either $j$ was already permanently evicted from $V_t$ at some time prior to $t_j$ (due to another distinguishing element), or $j \in V_{t_j}$. If $j \in V_{t_j}$, then $w_j \notin O_{t_j-1}$ and thus $w_j$ enters the sure set $S_{t_j}$. Consequently, $h_j$ is permanently ruled out from $V_t$ for all $t \ge t_j$. In either case, every $j \in B$ is evicted from $V_t$ by some finite time.

Let $t^\star \coloneqq \max\{t_j : j \in B\}$, with the convention $t^\star = z$ if $B = \emptyset$. Then, for all $t \ge t^\star$ and all $j < z$, if $S_t \subseteq \operatorname{supp}(h_j)$, necessarily $j \notin B$ and hence $\operatorname{supp}(h_z) \subseteq \operatorname{supp}(h_j) \cup O_{z-1} \subseteq \operatorname{supp}(h_j) \cup O_{t-1}$. Intersecting with $\{1, \ldots, m\}$ yields

$\mathrm{supp}\,(h_z)\,[m] \subseteq \mathrm{supp}\,(h_j)\,[m] \cup O_{t-1}$ for all $m \in \mathbb{N}$. Since we also have that $S_t \subseteq \mathrm{supp}\,(h_z)$ for all $t$, the hypothesis $h_z$ satisfies Definition B.1 for all $t \geq t^\star$ and $m \in \mathbb{N}$. $\square$

**Lemma B.3** (Per-round termination). *For every $t \in \mathbb{N}$, Algorithm 2 outputs $o_t$ after finitely many iterations of the repeat-until loop.*

*Proof.* Fix $t \in \mathbb{N}$. If $V_t = \emptyset$, the algorithm outputs some $o_t \in S_t$ and terminates. Assume $V_t \neq \emptyset$. Then, for each $m$, there exists at least one $(t, m)$-critical hypothesis: the minimal index in $V_t$ is $(t, m)$-critical with replay, since condition (ii) of Definition B.1 is vacuous in this case.

As $m$ increases, the predicate "$h_i$ is $(t, m)$-critical with replay" is monotone: once false for a certain $m'$, it remains false for all $m \geq m'$.[8] Thus, $n^{(t,m)}$ is nonincreasing in $m$. Additionally, it takes values in the finite set $\{1, \ldots, t\}$. Hence, there exists $m_0 < \infty$ such that $n^{(t,m)} = \bar{n}$ for all $m \geq m_0$.

Fix any $m \geq m_0$. The excluded set in the output-selection loop,

$$E^{(t,m)} := S_t \cup O_{t-1} \cup W^{(t,m)},$$

is finite and has size at most $|S_t| + |O_{t-1}| + |V_t|\,(|V_t| - 1)\,/2 \leq 2t + t^2$. Since $\mathrm{supp}\,(h_{\bar{n}})$ is infinite, the cardinality of $\mathrm{supp}\,(h_{\bar{n}})\,[m]$ diverges with $m$. Therefore, for all sufficiently large $m \geq m_0$ we have

$$\mathrm{supp}\,(h_{\bar{n}})\,[m] \setminus E^{(t,m)} \neq \emptyset.$$

For such an $m$, the for-loop finds an admissible $x$ and sets $o_t \neq \perp$, causing the repeat-until loop to terminate. $\square$

**Lemma B.4** (Eventual validity). *There exists $t^\star < \infty$ such that for all $t \geq t^\star$ the output satisfies*

$$o_t \in \mathrm{supp}\,(h_z) \setminus \{x_1, \ldots, x_t\}.$$

*Proof.* Let $t^\star$ be as in Lemma B.2 and fix $t \geq t^\star$. The branch $V_t = \emptyset$ cannot occur since $h_z$ is always consistent with $S_t$ and hence $z \in V_t$. Thus, $V_t \neq \emptyset$ and the algorithm outputs some $o_t \in \mathrm{supp}\,(h_{n^{(t,m)}})\,[m] \setminus \left(S_t \cup O_{t-1} \cup W^{(t,m)}\right)$ for the final value of $m$ in the repeat-until loop.

Since $h_z$ is $(t, m)$-critical, we have $n^{(t,m)} \geq z$. If $n^{(t,m)} = z$, then $o_t \in \mathrm{supp}\,(h_z)$ immediately. If $n^{(t,m)} > z$, applying condition (ii) of Definition B.1 to the pair $(i, n) = (z, n^{(t,m)})$ yields

$$\mathrm{supp}\,(h_{n^{(t,m)}})\,[m] \subseteq \mathrm{supp}\,(h_z)\,[m] \cup O_{t-1}.$$

Because $o_t \notin O_{t-1}$ by construction, it follows that $o_t \in \mathrm{supp}\,(h_z)$. Finally, $o_t \notin S_t$ and $o_t \notin O_{t-1}$ implies $o_t \notin \{x_1, \ldots, x_t\}$, since every observed example is either sure (hence in $S_t$) or a replay (hence in $O_{t-1}$). Thus, $o_t \in \mathrm{supp}\,(h_z) \setminus \{x_1, \ldots, x_t\}$, as claimed. $\square$

We can finally provide a proof of Theorem 6.1.

*Proof of Theorem 6.1.* Algorithm 2 only requires membership queries to evaluate $h_i(x)$ for $i \leq t$ and $x \leq m$, where $t, m$ are finite. Additionally, Lemma B.3 shows that, at every time $t$, Algorithm 2 outputs some $o_t$ after finitely many operations. Hence, Algorithm 2 is a computable procedure that can be implemented using membership queries alone. Now, fix any target $h^\star \in \mathcal{H}$. For any enumeration with replay for $h^\star$ and WP, Lemma B.4 gives a time $t^\star$ after which every output is fresh and valid for $h^\star$. Therefore, Algorithm 2 generates $h^\star$ in the limit with replay. Since $h^\star \in \mathcal{H}$ was arbitrary, the theorem follows. $\square$

---

[8]Indeed, condition 1 of Definition B.1 is independent of $m$. Moreover, if condition 2 fails at some prefix $m'$, then there exist $j < i$ with $S_t \subseteq \mathrm{supp}\,(h_j)$ and some $x \leq m'$ such that $x \in \mathrm{supp}\,(h_i)$ and $x \notin \mathrm{supp}\,(h_j) \cup O_{t-1}$. The same $x$ belongs to every larger prefix, so condition 2 also fails for every $m \geq m'$. Therefore, the set of $(t, m)$-critical hypotheses can only shrink as $m$ increases, and $n^{(t,m)}$ is nonincreasing in $m$.

## C. Proof of Theorem 6.2

We first restate the main theorem we are proving in this section.

**Theorem 6.2.** *There exists a hypothesis class $\mathcal{H}$ that is generatable in the limit but is not generatable in the limit with replay.*

We prove Theorem 6.2 by an explicit construction loosely based on Bai et al. (2026). First, we need to introduce some additional notation. Let the domain be $\mathcal{X} := \mathbb{Z} \cup \{*^n \mid n \in \mathbb{N}\}$. Strings of the form $*^n$ act as "special tokens" that will index the relevant subclass. For $b \in \mathbb{N}_0$, define

$$\mathcal{H}_1^b := \left\{ h \in \{0,1\}^{\mathcal{X}} \;\middle|\; \mathrm{supp}\,(h) = \{b\} \cup A \cup \{x \in \mathbb{Z} : x > j\} \text{ for some } A \subseteq \mathbb{Z}, \ j > b \right\},$$

$$\mathcal{H}_2^b := \left\{ h \in \{0,1\}^{\mathcal{X}} \;\middle|\; \mathrm{supp}\,(h) = \{x \in \mathbb{Z} : x < b\} \cup A \text{ for some } A \subseteq \mathbb{Z} \setminus \{b\} \right\},$$

and let $\mathcal{H}^b := \mathcal{H}_1^b \cup \mathcal{H}_2^b$. Informally, $\mathcal{H}_1^b$ contains hypotheses whose support includes $b$ and contains all integers larger than some cutoff $j > b$, whereas $\mathcal{H}_2^b$ contains hypotheses that omit $b$ but include every integer less than $b$. Both classes also include an arbitrary subset $A$ of the remaining integers.[9]

Next, for $i \in \{1, 2\}$ define

$$\widetilde{\mathcal{H}}_i^b := \left\{ \tilde{h} \in \{0,1\}^{\mathcal{X}} \;\middle|\; \mathrm{supp}\left(\tilde{h}\right) = \mathrm{supp}\,(h) \cup \{*^k : 1 \le k \le b\} \text{ for some } h \in \mathcal{H}_i^b \right\},$$

and let $\widetilde{\mathcal{H}}^b := \widetilde{\mathcal{H}}_1^b \cup \widetilde{\mathcal{H}}_2^b$. Thus, $\widetilde{\mathcal{H}}_i^b$ is obtained from $\mathcal{H}_i^b$ by adding the marker strings $*^1, \ldots, *^b$ to the support of each hypothesis. Finally, define the class

$$\mathcal{H} := \left\{ h^{\mathrm{mk}} \right\} \cup \bigcup_{b \in \mathbb{N}_0} \widetilde{\mathcal{H}}^b \quad \text{with} \quad h^{\mathrm{mk}} := \mathbf{1}\left\{ *^n \mid n \in \mathbb{N} \right\}.$$

That is, $\mathcal{H}$ contains the all-marker hypothesis $h^{\mathrm{mk}}$ together with all the padded classes $\widetilde{\mathcal{H}}^b$.

To prove Theorem 6.2, Lemma C.1 shows that $\mathcal{H}$ is generatable in the limit, while Lemma C.2 shows that $\mathcal{H}$ is not generatable in the limit with replay.

**Lemma C.1.** *The class $\mathcal{H}$ is generatable in the limit.*

*Proof.* Fix $b \in \mathbb{N}_0$. Let $\mathcal{G}^b$ be the generator from Bai et al. (2026) that generates $\mathcal{H}^b$ in the limit. We briefly recall its definition:

$$\mathcal{G}^b(x_1, \ldots, x_t) := \begin{cases} \max\{t, o_1, \ldots, o_{t-1}, x_1, \ldots, x_t\} + 1 & \text{if } b \in \{x_1, \ldots, x_t\}, \\ \min\{b, o_1, \ldots, o_{t-1}, x_1, \ldots, x_t\} - 1 & \text{otherwise.} \end{cases}$$

Essentially, if $h^\star \in \mathcal{H}_1^b$ then $\mathcal{G}^b$ will observe $b$ and eventually output unseen integers larger than the cutoff $j$; otherwise, if $h^\star \in \mathcal{H}_2^b$ then $\mathcal{G}^b$ will always take the second branch and output unseen integers smaller than $b$. Since $\widetilde{\mathcal{H}}^b$ only augments each $h \in \mathcal{H}^b$ by the same finite set of $*$-strings, the same generator $\mathcal{G}^b$ (ignoring the $*$-strings in its input) generates $\widetilde{\mathcal{H}}^b$ in the limit.

We now use this to define a single generator $\mathcal{G}$ for $\mathcal{H}$. Given a history $(x_1, \ldots, x_t)$, define

$$m(t) := \max\left\{ k \in \mathbb{N} : *^k \in \{x_1, \ldots, x_t\} \right\},$$

with $m(t) = 0$ if no $*$-string has appeared. Let $Z_t := (x_s : x_s \in \mathbb{Z}, \ 1 \le s \le t)$ denote the subsequence of integer-valued examples in $x_{1:t}$, and set

$$\mathcal{G}(x_1, \ldots, x_t) := \begin{cases} *^{m(t)+1} & \text{if } \{x_1, \ldots, x_t\} \subseteq \{*^n \mid n \in \mathbb{N}\}, \\ \mathcal{G}^{m(t)}(Z_t) & \text{otherwise.} \end{cases}$$

---

[9]We note that $\mathcal{H}^b$ is generatable in the limit but not generatable in the limit with a single omission (i.e., by omitting $b$), as shown by Bai et al. (2026) (where they set $b = 0$). However, $\mathcal{H}^b$ is generatable in the limit with replay: the generator of Bai et al. (2026) that works in the standard setting can be adapted to the replay setting by restricting it from outputting the crucial string $b$. Therefore, in order to show a separation between generation in the limit with and without replay, we will need a more involved construction.

Thus, if $h^\star = \mathbf{1}\{*^n \mid n \in \mathbb{N}\}$ then the output of $\mathcal{G}$ is always an unseen $*$-string, so $\mathcal{G}$ generates $h^\star$ in the limit. Otherwise, for any $h^\star \in \bigcup_b \widetilde{\mathcal{H}}^b$, note that $b^\star := \max\{k : *^k \in \mathrm{supp}(h^\star)\}$ equals the unique index such that $h^\star \in \widetilde{\mathcal{H}}^{b^\star}$. Hence, on any enumeration $(x_t)_{t \geq 1}$ of $\mathrm{supp}(h^\star)$, the value of $m(t)$ is nondecreasing and stabilizes to $b^\star$ after the finite time $t'$ when $*^{b^\star}$ appears in the enumeration. Moreover, since $\mathrm{supp}(h^\star) \cap \mathbb{Z}$ is infinite, any enumeration of $\mathrm{supp}(h^\star)$ must present an integer at some finite time $t''$; after that time, $\mathcal{G}$ always takes the second branch. Therefore, $\mathcal{G}$ copies $\mathcal{G}^{b^\star}$ for all $t \geq \tilde{t} := \max\{t', t''\}$. Since $\mathcal{G}^{b^\star}$ generates $\widetilde{\mathcal{H}}^{b^\star}$ in the limit, there exists $t^\star \in \mathbb{N}$ such that $\mathcal{G}^{b^\star}(Z_t) \in \mathrm{supp}(h^\star) \setminus \{x_1, \ldots, x_t\}$ for all $t \geq t^\star$. Thus, $\mathcal{G}(x_1, \ldots, x_t)$ is guaranteed to be a valid output for all $t \geq \max\{\tilde{t}, t^\star\}$. $\square$

**Lemma C.2.** *The class $\mathcal{H}$ is not generatable in the limit with replay.*

*Proof.* Assume for the sake of contradiction that there exists a generator $\mathcal{G}$ that generates $\mathcal{H}$ in the limit with replay. We construct an adversarial enumeration with replay $(x_t)_{t \geq 1}$ adaptively. Since $\mathcal{G}$ is deterministic, the adversary can choose the next input based on $\mathcal{G}$'s outputs while maintaining a nonempty set of candidate target hypotheses consistent with the observed prefix stream. The construction identifies a hypothesis $h^\star \in \mathcal{H}$ that remains consistent with the entire realized stream $(x_t)_{t \geq 1}$ and for which $\mathcal{G}$ outputs infinitely many invalid elements. Write $o_t := \mathcal{G}(x_{1:t})$. The enumeration is defined in two steps.

In the first step, the adversary forces the generator to output a "long" $*-$string marker. Let $h^{\mathrm{mk}} := \mathbf{1}\{*^n \mid n \in \mathbb{N}\}$. The adversary begins enumerating the support of $h^{\mathrm{mk}}$ by presenting the sequence $x_t := *^t$. Since $\mathcal{G}$ generates $\mathcal{H}$ in the limit with replay, there exists a time step $\tau$ such that $o_\tau \in \mathrm{supp}(h^{\mathrm{mk}}) \setminus \{x_1, \ldots, x_\tau\}$. Let the output be $o_\tau = *^z$ for some $z > \tau$. The adversary then extends the input sequence by displaying all $*-$strings until $*^z$ by setting $x_{\tau+1} := *^{\tau+1}, \ldots, x_z := *^z$. Let $J_0 := z$.[10]

In the second step, the adversary forces infinitely many mistakes in multiple phases. For each $n \in \mathbb{N}$, at the beginning of phase $n$, the adversary presents the integer $z - n$, and then presents the increasing tail $J_{n-1} + 1, J_{n-1} + 2, \ldots$ until the first time $t_n$ at which $\mathcal{G}$ outputs an integer $o_{t_n}$ satisfying $o_{t_n} > J_{n-1}$ and $o_{t_n} \notin \{x_1, \ldots, x_{t_n}\}$. The adversary then sets $J_n := o_{t_n}$ and proceeds to phase $n + 1$, never presenting $J_n$ as an element of the sequence.

*Claim* 1. Each phase terminates, i.e., $t_n < \infty$ for all $n \in \mathbb{N}$.

We prove this claim later. First, we show how the adversary can use this fact to force infinitely many mistakes. Let $S \subseteq \mathbb{Z}$ be the set of all integers ever presented by the above construction during the second step, and define

$$\mathrm{supp}(h^\star) := S \cup \{*^k : 1 \leq k \leq z\} = \bigcup_{t \geq 1} \{x_t\}.$$

Since *every* integer smaller than $z$ appears in the enumeration $(x_t)_{t \geq 1}$ as $z - n$ for some $n \in \mathbb{N}$, we can write

$$S = \{x \in \mathbb{Z} : x < z\} \cup A \quad \text{where} \quad A := \{x \in S : x > z\} \subseteq \mathbb{Z} \setminus \{z\}.$$

This shows that the example sequence $(x_t)_{t \geq 1}$ enumerates the support of a hypothesis $h^\star$ that belongs to $\widetilde{\mathcal{H}}_2^z$. However, by construction, $o_{t_n} = J_n \notin \mathrm{supp}(h^\star)$ for each $n \in \mathbb{N}$. Hence, $(t_n)_{n \geq 1}$ is an infinite sequence of time steps at which $\mathcal{G}$ makes a mistake, contradicting that $\mathcal{G}$ generates $\mathcal{H}$ in the limit with replay. $\square$

*Proof of Claim 1.* Fix $n \in \mathbb{N}$. Suppose, for the sake of contradiction, that phase $n$ does not terminate, i.e., the adversary keeps presenting $J_{n-1} + 1, J_{n-1} + 2, \ldots$, but the generator never outputs a fresh integer larger than $J_{n-1}$. Let $\hat{h}$ be the hypothesis whose support is enumerated by such an example sequence, excluding the (potentially replayed) string $*^z$. Denote by $X_{<n} \subseteq \mathbb{Z}$ the finite set of integers that appear *before* the beginning of phase $n$, with $X_{<1} = \emptyset$ for $n = 1$. Then, we can write

$$\mathrm{supp}(\hat{h}) := X_{<n} \cup \{z - n\} \cup \{x \in \mathbb{Z} : x > J_{n-1}\} \cup \{*^k : 1 \leq k \leq z - 1\}.$$

Notice that $z - 1$ belongs to the support of $\hat{h}$ for any $n \in \mathbb{N}$: for $n = 1$, it appears directly as $z - n$; for all other $n$, it already appeared during a previous phase and is contained in $X_{<n}$. Consequently, $\hat{h}$ is a valid hypothesis from $\widetilde{\mathcal{H}}_1^{z-1}$. By

---

[10]At a high level, note that since we are in the replay setting, $\mathcal{G}$ cannot know whether $*^z$ belongs to the support of the target hypothesis $h^\star$ or not. Hence, even upon observing $z - 1$, $\mathcal{G}$ does not know whether $h^\star \in \widetilde{\mathcal{H}}^{z-1}$—in which case $h^\star \in \widetilde{\mathcal{H}}_1^{z-1}$ and $\mathrm{supp}(h^\star)$ necessarily contains all integers larger than some cutoff $j$—or $h^\star \in \widetilde{\mathcal{H}}^z$, in which case observing $z - 1$ is uninformative, as $h^\star$ could belong to either $\widetilde{\mathcal{H}}_1^z$ or $\widetilde{\mathcal{H}}_2^z$. The second step of our construction relies on this observation to force infinitely many mistakes.

construction, the only string in the adversary's sequence that does not belong to the support of $\hat{h}$ is $*^z$, which nevertheless appears in the sequence as a replay of the earlier output $o_\tau = *^z$. Thus, the adversary's sequence is a valid enumeration with replay for $\hat{h}$ and $\mathcal{G}$. Therefore, since $\mathcal{G}$ generates $\mathcal{H}$ in the limit with replay, $\mathcal{G}$ must eventually output an unseen element from the support of $\hat{h}$. Since all elements of $X_{<n} \cup \{z - n\} \cup \{*^k : 1 \leq k \leq z - 1\}$ have already appeared, any such fresh element must belong to the tail $\{x \in \mathbb{Z} : x > J_{n-1}\}$. Consequently, $\mathcal{G}$ outputs an unseen integer larger than $J_{n-1}$ at a finite time, contradicting the assumption that phase $n$ does not terminate. $\square$

*Proof of Theorem 6.2.* The hypothesis class $\mathcal{H}$ is generatable in the limit (Lemma C.1), but not generatable in the limit with replay (Lemma C.2). $\square$

# D. Proof of Theorem 7.1

We first restate the main theorem we are proving in this section.

**Theorem 7.1.** *There cannot exist a (deterministic) generator $\mathcal{G}$ that only makes membership queries and properly generates in the limit all countable hypothesis classes.*

As described in Algorithm 3, for any given computable proper generator $\mathcal{G}$ that only makes membership queries, we construct a hard class $\mathcal{H}$ on which $\mathcal{G}$ makes infinitely many mistakes by simulating $\mathcal{G}$'s interaction with an adversarial enumeration. At a high level, this follows the same "simulation template" as the computational lower bound of Charikar & Pabbaraju (2025); our construction, however, maintains a *countably infinite* class rather than only two hypotheses. Algorithm 3 defines $\mathcal{H} = \{h_1, h_2, \ldots\}$ via a function $F : \mathbb{N} \times \mathbb{N} \to \{0, 1\}$ defined as

$$F(i, j) = \begin{cases} 1 & \text{if } j \in \text{supp}(h_i), \\ 0 & \text{if } j \notin \text{supp}(h_i), \end{cases}$$

which constitutes the (limited) interface available to $\mathcal{G}$ to interact with the hypothesis class $\mathcal{H}$. To compute $F(i, j)$, one would run Algorithm 3—which in turn simulates $\mathcal{G}$—until the value of $F(i, j)$ is assigned.

Because $\mathcal{H}$ is countable, we can assume that $\mathcal{G}$ outputs an index $i_t \in \mathbb{N}$, interpreted as the index of the output hypothesis; that is, $\hat{h}_t = h_{i_t}$. Since $\mathcal{G}$ is restricted to membership queries, at every step $t$, it will have gathered information about finitely many hypotheses and finitely many instances (i.e., elements of the domain $\mathcal{X}$). Algorithm 3 maintains two counters $I$ and $J$ that delimit the finite "rectangle" of hypothesis-instance pairs $(i, j) \in \mathbb{N} \times \mathbb{N}$ queried so far by $\mathcal{G}$; outside this rectangle, it sets memberships adversarially. To ensure that the revealed sequence enumerates the target support, the construction maintains a queue $Q$ and at each round reveals $x_t = \min Q$, which guarantees that each element entering $Q$ will be revealed after a finite number of rounds. Additionally, it maintains a *trap* pair $(i', j')$ of hypothesis $h_{i'}$ and instance $j'$ such that $j' \notin \text{supp}(h_{i'})$ but $j' \in \text{supp}(h_i)$ for all $i \neq i'$. The hypothesis $h_1$ serves as a reference hypothesis.

The algorithm has two modes—*diagonalization* and *overgeneralization*—and it switches mode automatically by adapting the enumeration $(x_t)_{t \geq 1}$ to $\mathcal{G}$'s outputs, specifically to whether $\hat{h}_t = h_1$. The current trap instance enters the enumeration queue $Q$ only at the first subsequent round $t$ (if ever) for which $\hat{h}_t \neq h_1$. When this occurs, Algorithm 3 also instantiates a new trap pair $(i', j')$ with $i' > I$ and $j' > J$. There are two cases:

- $\mathcal{G}$ outputs a hypothesis different from $h_1$ *infinitely* often. In this case, the adversary enumerates $\text{supp}(h_1)$ and forces $\mathcal{G}$ to make infinitely many mistakes via *diagonalization*: at each round $t$ with $\hat{h}_t \neq h_1$, it inserts a fresh instance $d_t$ beyond the counter $J$ and assigns it to the support of $\hat{h}_t$ but not to that of $h_1$.

- $\mathcal{G}$ outputs a hypothesis different from $h_1$ only *finitely* often. Then, after some finite time, it outputs $h_1$ indefinitely. In this case, the adversary enumerates the support of the current *trap* hypothesis $h_{i'}$, whose support is strictly smaller than $\text{supp}(h_1)$, so that $\mathcal{G}$ *overgeneralizes*.

Figure 1 illustrates a few steps of this procedure.

To prove Theorem 7.1, we first argue about the soundness of our construction by analyzing the function $F$, showing that the corresponding hypothesis class $\mathcal{H}$ is an *indexed family of recursive languages* (in the sense of Angluin (1980)) satisfying the UUS assumption.

---

**Algorithm 3** Hard Hypothesis Class for the Proper Generator $\mathcal{G}$

---

**Require:** Proper generator $\mathcal{G}$
1: Set $F(i,1) = 1$ for all $i \in \mathbb{N}$
2: Initialize enumeration queue: $Q \leftarrow \{1\}$
3: Set up the trap: $F(i,2) = \begin{cases} 0 & \text{if } i = 2, \\ 1 & \text{if } i \neq 2 \end{cases}$
4: Initialize trap pair $(i', j') \leftarrow (2,2)$
5: Initialize counters: $I \leftarrow 2$ and $J \leftarrow 2$
6: **for** t=1,2,... **do**
7:     Show $\mathcal{G}$ the example $x_t \leftarrow \min Q$; Remove $x_t$ from $Q$
8:     $k \leftarrow 1$
9:     **while** $\mathcal{G}$ issues a new membership query $(i,j)$ **do**
10:         $m \leftarrow \max\{j,k\}$
11:         **if** $m > J$ **then**
12:             **for** $n = J+1, J+2, \ldots, m$ **do**
13:                 Set $F(\ell, n) = 1$ for all $\ell \in \mathbb{N}$; Add $n$ to $Q$
14:             **end for**
15:         $J \leftarrow m$
16:         **end if**
17:         $I \leftarrow \max\{I, i\}$
18:         $k \leftarrow k + 1$
19:     **end while**
20:     Receive $\mathcal{G}$'s output $i_t$ {Interpreted as $\hat{h}_t = h_{i_t}$}
21:     $I \leftarrow \max\{I, i_t\}$
22:     **if** $i_t \neq 1$ **then**
23:         Add $j'$ to $Q$
24:         Diagonalization step: $d_t \leftarrow J+1$; Set $F(i, d_t) = \begin{cases} 1 & \text{if } i = i_t, \\ 0 & \text{if } i \neq i_t \end{cases}$
25:         Set up a new trap: $e_t \leftarrow J+2$; Set $F(i, e_t) = \begin{cases} 0 & \text{if } i = I+1, \\ 1 & \text{if } i \neq I+1 \end{cases}$
26:         Update trap pair: $(i', j') \leftarrow (I+1, e_t)$
27:         Update counters: $I \leftarrow i'$ and $J \leftarrow e_t$
28:     **end if**
29:     Let $c_t \leftarrow J+1$; Set $F(i, c_t) = 1 \; \forall i \in \mathbb{N}$; Add $c_t$ to $Q$; Update $J \leftarrow c_t$
30: **end for**

---

**Lemma D.1.** *For any computable $\mathcal{G}$, the associated function $F : \mathbb{N} \times \mathbb{N} \to \{0,1\}$ defined in Algorithm 3 is total recursive. Moreover, for every $i \in \mathbb{N}$, the set $\{j \in \mathbb{N} \mid F(i,j) = 1\}$ is infinite.*

*Proof.* We show that, for every pair $(i,j) \in \mathbb{N} \times \mathbb{N}$, the value $F(i,j)$ is decided at a finite step of Algorithm 3 and is never changed afterward. Observe that whenever Algorithm 3 encounters an instance $j$, it assigns the entire row $F(\cdot, j)$ in a single step, i.e., it fixes $F(i,j)$ for all $i \in \mathbb{N}$ at once. Therefore, it suffices to show that every instance $j \in \mathbb{N}$ is encountered exactly once and at a finite step. To this end, we claim that, throughout the execution of Algorithm 3, the set of encountered instances is always exactly the initial segment $\{1, \ldots, J\}$, meaning that no instance $j < J$ is skipped during the execution. This is true at initialization: the algorithm encounters instances 1 and 2, and then sets $J \leftarrow 2$. Now consider any later stage of the construction. During the processing of $\mathcal{G}$'s membership queries, suppose that $\mathcal{G}$ issues its $k$-th query $(i,j)$ in the current round. Then, Algorithm 3 assigns values to all still-unseen instances $n$ satisfying

$$J < n \leq \max\{j, k\},$$

and updates $J$ accordingly. Hence, all newly encountered instances form a consecutive block immediately after the current value of $J$. In particular, no instance is skipped, and no previously encountered instance is revisited or modified. In all other

---

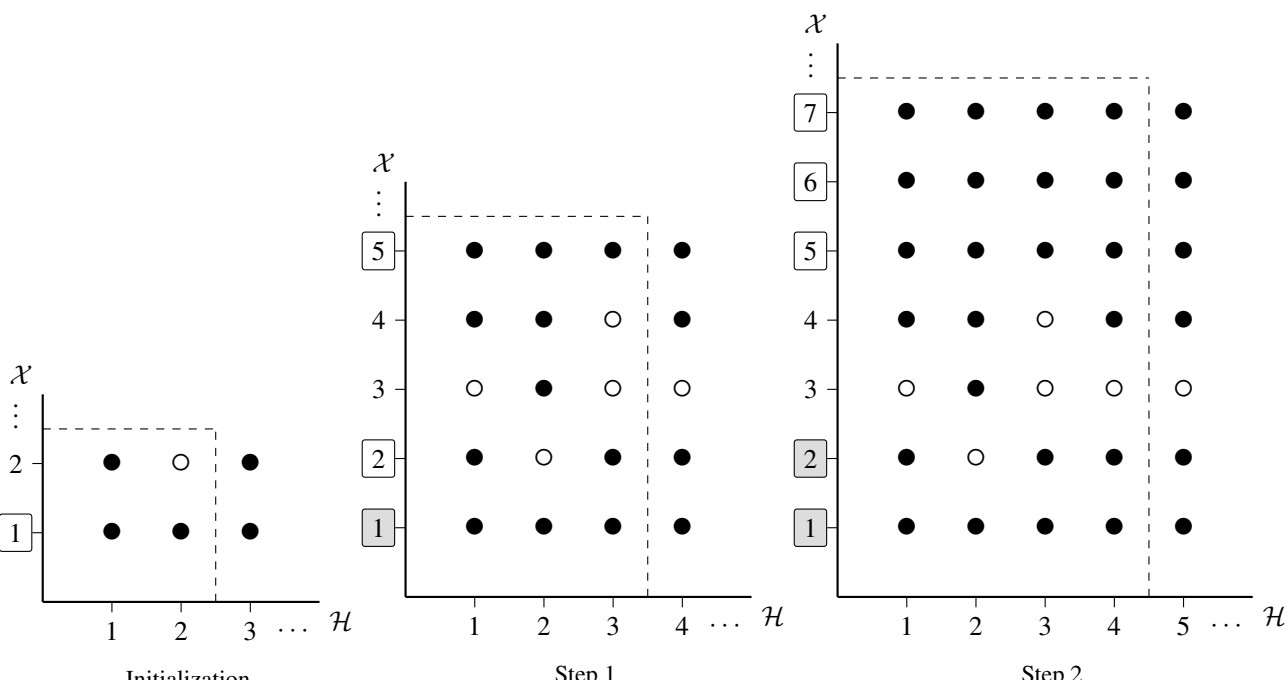

*Figure 1.* Online construction of a hard hypothesis class for a given proper generator.

The horizontal axis represents the hypotheses in $\mathcal{H}$, and the vertical axis represents the instances from the domain $\mathcal{X}$. For every coordinate pair $(i, j)$, a filled circle ($\bullet$) indicates $j \in \mathrm{supp}\,(h_i)$, while an empty circle ($\circ$) indicates $j \notin \mathrm{supp}\,(h_i)$. A box around a label on the vertical axis means that the instance has been added to the enumeration queue $Q$, while a shaded box means that the instance has been shown as an example $x_t$. Finally, the L-shaped dashed line marks the current boundaries of $\mathcal{G}$'s knowledge, as tracked by $I$ and $J$.

We illustrate the first steps of the interaction. At initialization, the adversary inserts instance 1 into the enumeration queue $Q$ and installs the *trap* hypothesis-instance pair $(i', j') = (2, 2)$. The counters $I$ and $J$ are both set to 2. At step 1, the adversary reveals $x_1 = 1$. For illustrative purposes, we assume that at step 1 the generator $\mathcal{G}$ outputs $\hat{h}_1 = h_2$. This triggers the *diagonalization* mode of Algorithm 3: instance $d_1 = 3$ is assigned exclusively to the output hypothesis $h_2$; the current trap instance $j' = 2$ is added to $Q$; a new trap hypothesis-instance pair $(i', j') = (3, 4)$ is created beyond $I$ and $J$ by assigning instance $e_1 = 4$ to all hypotheses except for $h_3$; finally, instance $c_1 = 5$ is assigned to all hypotheses and is therefore added to the enumeration queue. When the round ends, the counters $I$ and $J$ are set to 3 and 5, respectively. Then step 2 begins with $x_2 = \min Q = 2$ being revealed to $\mathcal{G}$. We assume that $\mathcal{G}$ queries $F(4, 6)$: instance 6 is therefore assigned to all hypotheses and added to $Q$. Furthermore, the counters $I$ and $J$ move to 4 and 6, respectively. Suppose $\mathcal{G}$ outputs $\hat{h}_2 = h_1$. This time the *overgeneralization* mode of Algorithm 3 is triggered. In this case, the trap hypothesis-instance pair remains the same. At the end of the round, $c_2 = 7$ is added to $Q$ and the counter $J$ is updated to 7.

steps where the algorithm introduces new instances, it uses fresh indices immediately following the current counter (i.e., $J + 1, J + 2$) and updates $J$ immediately afterward. Thus, the set of encountered instances remains an initial segment of $\mathbb{N}$ at every stage.

It remains to show that every instance is encountered at a finite step, or equivalently, that the counter $J$ is unbounded. There are two cases. First, suppose that in every round $t$, the generator $\mathcal{G}$ asks only finitely many membership queries and eventually outputs some $\hat{h}_t$. Then, each iteration of the outer loop completes, and the last line of Algorithm 3 introduces a fresh instance $c_t$ at the end of every round (even if $\mathcal{G}$ asks no query in that round). Therefore, $J$ increases by at least one in every round, so $J \to \infty$ as $t \to \infty$. Second, suppose that in some round, $\mathcal{G}$ asks infinitely many membership queries and never produces an output.[11] Let $k$ denote the query counter within that round and note that, by construction, $J \geq k$. Since $k$ is unbounded along that infinite query sequence, $J$ is also unbounded within that single round. Hence, in either case, $F$ is recursive over its whole domain.

Finally, the same two-case analysis shows that the resulting hypothesis class satisfies the UUS property. If every round is finite, then the fresh instance $c_t$ introduced at the end of each round is assigned to all hypotheses, so each support

---

[11] For instance, suppose $\mathcal{G}$ keeps querying whether $j \in \mathrm{supp}\,(h_i)$ for a fixed $j$ and different $i$ until it gets a negative answer. Clearly, in this case $\mathcal{G}$ would fail at its generation task, granted of course that the hypothesis class resulting from the construction was still valid.

supp $(h_i)$ contains infinitely many such instances. If instead some round contains infinitely many queries, then as $k \to \infty$, the construction introduces infinitely many new instances during that round, and each of them is again assigned to all hypotheses. Thus, supp $(h_i)$ is infinite for all $i \in \mathbb{N}$ in either case. □

Having shown that Algorithm 3 defines a valid hypothesis class, we can now provide the proof of Theorem 7.1.

*Proof of Theorem 7.1.* Suppose, for the sake of contradiction, that there exists a deterministic proper generator $\mathcal{G}$ that uses only membership queries and properly generates in the limit every countable hypothesis class. Consider the hypothesis class $\mathcal{H} = \{h_1, h_2, \ldots\}$ induced by Algorithm 3 when run against $\mathcal{G}$. Since $\mathcal{G}$ properly generates $\mathcal{H}$, we can assume that at each step $t$ the generator $\mathcal{G}$ halts to produce an output hypothesis $\hat{h}_t = h_{i_t} \in \mathcal{H}$. For every round $t$ with $i_t \neq 1$, let $d_t$ denote the fresh diagonalization instance created at round $t$ by Algorithm 3. By construction,

$$d_t \in \text{supp}(h_{i_t}) \quad \text{and} \quad d_t \notin \text{supp}(h_i) \;\; \forall i \neq i_t,$$

so in particular $d_t \notin \text{supp}(h_1)$. Also, if $(i', j')$ denotes the current trap pair maintained by the algorithm, then it holds that

$$j' \notin \text{supp}(h_{i'}) \quad \text{and} \quad j' \in \text{supp}(h_i) \;\; \forall i \neq i'.$$

Additionally, define

$$Q_\infty := \{n \in \mathbb{N} : n \text{ is ever added to } Q\}.$$

Since at each round $t$ the algorithm reveals $x_t = \min Q$, every $n \in Q_\infty$ is shown after finitely many rounds: only finitely many smaller integers can ever be inserted before it, and each of them is removed after one round. Therefore, $(x_t)_{t \geq 1}$ is an enumeration of $Q_\infty$. We now distinguish two cases: $\mathcal{G}$ outputs $\hat{h}_t \neq h_1$ either finitely or infinitely many times.

First, suppose $\mathcal{G}$ outputs $\hat{h}_t \neq h_1$ *infinitely* many times.[12] We argue that this would lead to a contradiction by showing that in this case Algorithm 3 enumerates supp $(h_1)$ and that $\mathcal{G}$ makes infinitely many mistakes for $h^\star = h_1$. To begin, note that if $h^\star = h_1$ then $\mathcal{G}$ makes infinitely many mistakes, since each $d_t$ only belongs to the support of $\hat{h}_t$ and thus

$$\text{supp}\left(\hat{h}_t\right) \nsubseteq \text{supp}(h_1).$$

It remains to show that, in this case, $Q_\infty = \text{supp}(h_1)$. Observe that all instances that are not trap instances are immediately added to $Q$ after being encountered. Additionally, any trap instance $e_t$ created at round $t$ is added to $Q$ at the next round $s > t$ such that $\mathcal{G}$ outputs $\hat{h}_s \neq h_1$, which we have assumed to be happening infinitely often in this case. Conversely, the only instances never added to $Q$ are the diagonalization instances $d_t$, and none of them belongs to supp $(h_1)$.

Therefore, it must be that $\mathcal{G}$ outputs $\hat{h}_t \neq h_1$ only *finitely* many times. Let

$$t_0 := \max\{t \in \mathbb{N} : i_t \neq 1\},$$

with the convention $t_0 = 0$ if $i_t = 1$ for all $t \in \mathbb{N}$. Let $(\bar{\imath}, \bar{\jmath})$ denote the values of the trap pair $(i', j')$ after round $t_0$; if $t_0 = 0$, then $(\bar{\imath}, \bar{\jmath}) = (2, 2)$. By definition of $t_0$, $\hat{h}_t = h_1$ for all $t > t_0$. We claim that $Q_\infty = \text{supp}(h_{\bar{\imath}})$. We first show that $Q_\infty \subseteq \text{supp}(h_{\bar{\imath}})$. Any instance introduced during the query phase or as some $c_t$ belongs to all hypotheses, hence in particular to $h_{\bar{\imath}}$. Any trap instance that is ever released and added to $Q$ must have been created before round $t_0$. If it was created when the trap index was $\tilde{\imath}$, then it is excluded only from $h_{\tilde{\imath}}$; since trap indices are strictly increasing, we have $\tilde{\imath} \neq \bar{\imath}$, and thus this instance also lies in supp $(h_{\bar{\imath}})$. Therefore, $Q_\infty \subseteq \text{supp}(h_{\bar{\imath}})$. Conversely, the only instances never added to $Q$ are precisely the final trap $\bar{\jmath}$ and the diagonalization instances $d_t$ created at rounds $t \leq t_0$ with $i_t \neq 1$. By definition, $\bar{\jmath} \notin \text{supp}(h_{\bar{\imath}})$. Moreover, each such $d_t$ belongs only to $h_{i_t}$, whereas the trap created in round $t$ has index strictly larger than $i_t$; since trap indices only increase afterward, $\bar{\imath} > i_t$, so $d_t \notin \text{supp}(h_{\bar{\imath}})$. Thus, supp $(h_{\bar{\imath}}) \subseteq Q_\infty$. We conclude that $Q_\infty = \text{supp}(h_{\bar{\imath}})$, so $(x_t)_{t \geq 1}$ is an enumeration of supp $(h_{\bar{\imath}})$ and we can set $h^\star = h_{\bar{\imath}}$. However, this implies that $\mathcal{G}$ makes infinitely many mistakes also in this case, since $\bar{\jmath} \in \text{supp}(h_1)$ but $\bar{\jmath} \notin \text{supp}(h_{\bar{\imath}})$.

As both cases yield a contradiction, we conclude that no deterministic generator using only membership queries can properly generate in the limit all countable hypothesis classes. □

---

[12]The most natural choice would be $\hat{h}_t = h_{i'}$ since at each step, the trap hypothesis $h_{i'}$ is, in some sense, the minimal consistent hypothesis.

