# OpenReview forum: "Language Generation with Replay: A Learning-Theoretic View of Model Collapse"
_ICML.cc/2026/Conference — ICML 2026 regular_

### Official Review · Reviewer_aCSo · 2026-03-09

**Soundness:** 3
**Presentation:** 3
**Significance:** 3
**Originality:** 4
**Overall Recommendation:** 3
**Confidence:** 4

**Summary:**

The work extends the model of Kleinberg & Mullainathan for *language generation in the limit*. Specifically, they explore the notion of replay within this framework, a phenomenon known to lead to training data contamination and model collapse empirically. Under various hypothesis classes and generation frameworks, the authors showcases contexts in which generation with replay is no more challenging than without--as well as instances where there is a provable separation in complexity. These theoretical results are the product of generation algorithms which reduce to the optima for the setting without replay, as well as hard examples instance streams.

**Compliance With Llm Reviewing Policy:**

Affirmed.

**Key Questions For Authors:**

Are my noted bugs in the proofs truly bugs? If so, can they easily be fixed? (I am happy to increase my score if these are addressed).

What do these results suggest (if anything) about known scaling laws for language models?

**Limitations:**

The authors well note the limitations of their modeling and problem settings in the final section.

**Strengths And Weaknesses:**

# Strengths
The paper introduces an important and natural extension on the theoretical models of language generation. Model collapse is a known empirical phenomena with potentially grave implications for training larger models, and while many heuristics exist for mitigating this, there is a lack of theoretical justifications.

The majority of the theorems appear correct (apart from potential proof bugs noted below), and naturally refine the original model problem. The formalization of "replay" is well-constructed and intuitive. Moreover, the results agree with intuition and the proofs are non-trivial.

# Weaknesses
The main weaknesses are in terms of the paper's writing, and potential bugs (or at least ill-specified steps) in the proofs for theoretical results in Sections 5 and 6.

With respect to the writing, I find the paper is hard to parse without having previously studied the Kleinberg paper. I personally had to skim that work to better understand the problem, definitions, and nature of the results being proven. The current paper reads well for someone initiated on the new literature for this problem but is far from self-contained. I would encourage the authors to consider revising the structure to better introduce the problem (with and without replay), and furthermore provide more intuitions for the results and their proofs prior to the later formalizations. In fact, it is not revealed what is explicitly proven until around the fourth page of the paper (half way through). The abstract for example notes that we are looking at separations between the model with and without replay but does not provide a sharp picture.

With respect to potential proof bugs:
- In Theorem 5.1, the proof of Lemma B.2. invokes the monotonicity of “$h_i$ is $(t,m)$-critical with replay” which is seemingly true but should be further formalized / justified since the bulk of this lemma relies on this fact.
- For Theorem 5.3, the proof of Lemma C.1 appears correct but I think the adversaries power is not well explained. Is the adversary constructing an input stream subject to a fixed $h^\star$ or constructing the input stream *then* selecting a consistent target?
- Theorem 6.2 is potentially incorrect as written. The chosen adverarial sequence of $0, -1, -2$ followed by all positive integers is claimed to be an enumeration with replay for $h_1^+$ and $h_2^+$ but neither $-1$ nor $-2$ is in the support of these hypotheses. These are also not possible as replay unless the generator output a hypothesis containing them. Thus, it seems that the impossibility proof here fails in its current form (though it can probably be fixed with a different construction).


### Nitpicks:
- In Definition 2.4: "eventually reveals every $x \in supp(h)$ would be more precise as "for every $x \in supp(h)$, there exists $t$ with $x_t = x$ since replayed items can be replayed infinitely.

---

> ### Author Rebuttal · Authors · 2026-03-30
>
> We thank the reviewer for the very careful reading.  We agree that the highlighted proof steps are a bit too compressed and we appreciate the opportunity to make them explicit. We believe the issues raised are exposition issues rather than errors, and we will revise the paper as below.
>
> **Lemma B.2.** For fixed $t$ and $i$, the predicate ``$h_i$ is $(t,m)$-critical with replay’’ is  monotone in $m$: once it fails at  some $m’$, it remains false for all $m \geq m’$. The reason is that, condition (i) of Definition 5.2 does not depend on $m$. Moreover, if condition (ii) fails at some $m’$ then there exists $j<i$ and $x\leq m’$ such that $x \in supp(h_i)$ but $x \notin supp(h_j) \cup O_{t-1}$; the same witness $x$ will make condition (ii) fail for every $m \geq m’$. Hence, the set of $(t,m)$-critical hypotheses can only shrink as $m$ increases, and therefore the selected index $n^{(t,m)}$ (defined as the largest $(t,m)$-critical index in $V_t$) is nonincreasing in $m$. We will make this explicit in the proof of Lemma B.2.
>
> **Lemma C.2.** The adversary is not intended to choose a stream first and then retroactively change the target mid-construction. The argument works as: the adversary adaptively constructs a stream while maintaining a nonempty set of candidate feasible target hypotheses consistent with the observed prefix stream. As the generator is deterministic, the adversary can use the generator’s outputs to keep the stream on a "bad branch" and force infinitely many mistakes. At the end, a fixed $h^\star$ remains consistent with the entire realized stream and also witnesses the failure. Since the theorem is a worst case lower bound, exhibiting such a fixed $h^\star$ together with a consistent input sequence suffices. We agree this should be stated explicitly and will add a clarifying sentence near Lemma C.2. We also note here that this style of argument is consistent with diagonalization proofs [BPZ26, HKMV25] and lower bounds in online learning in general.
>
> **Theorem 6.2.** We agree that the highlighted part of the proof of Theorem 6.2 is indeed a bit terse and can be confusing. We will revise it to make the supports and the replay-validity more explicit.
>
> Let $ \hat{h}_t $ be the output of $ \mathcal{G} $ at $ t $.
>
> Recall that, in the *proper* setting, a sequence $(x_t)_{t\geq 1}$ is a valid *enumeration with replay* for $\mathcal{G}$ and $h$ if:
> 1. for all $t$, $x_t \in supp(h)$ or $x_t \in supp(\hat{h}_s)$ with $s < t$;
> 2. for every $x\in$ $supp(h)$ there exists $t$ such that $x=x_t$.
>
> The proof begins with the adversary revealing $x_1=0$. We condition on the case $\hat{h}_1 = h_1^{-}$ and note that the other cases are symmetric.
>
> Since the support of $\hat{h}_{1} = h_1^{-}$ contains all negative integers, both $x_2 = -1$ and $x_3 = -2$ are valid replay elements, as they appear after $h_1^{-}$ has been output by the generator.
>
> Therefore, the sequence $(x_t)_{t\geq 1}$ obtained by extending $0,-1,-2$ with all positive integers satisfies condition 1 for both $h_1^{+}$ and $h_2^{+}$: for $h_1^{+}$, we have $0, -1, 1, 2, 3,\ldots \in$ $supp(h_1^{+})$ while $-2\in \hat{h}_1$ is valid as replay; the case of $h_2^{+}$ is analogous.
>
> Furthermore, $(x_t)_{t\geq 1}$ contains an enumeration of the support of both $h_1^{+}$ and $h_2^{+}$, and thus also satisfies condition 2. We agree that this should be stated explicitly and will expand the proof accordingly.
>
> ---
>
> We thank the reviewer for raising these technical questions and hope our answers resolve their concerns. If not, we are happy to provide any additional clarification.
>
> ---
>
> **Scaling laws.** We also thank the reviewer for raising this interesting question. Our theorems mainly characterize whether replay preserves generatability at all, rather than how the error decays as a function of corpus size. As our results indicate, the first question is already interesting and produces several points of separation from generatability without replay. We believe this suggests that replay can change the effective learning curve and consequently impact the scaling laws, but making that connection precise is outside the scope of the current work and would require a more quantitative theory of rates in the replay setting. We will clarify this discussion in the revision and mention it as an interesting direction for future work.
>
> **Presentation.** We realized this may be hard to parse for readers unfamiliar with the literature. For that very reason, we made Section 2 large, providing a brief overview of the results before turning to the technical details. If the paper is accepted and the reviewer recommends it, we will use the additional camera-ready page to expand this section and make the paper more self-contained.
>
> ---
>
> [BPZ26] Bai, Y., Panigrahi, D. and Zhang, I. Language generation in the limit: Noise, loss, and feedback. SODA, 2026.
>
> [HKMV25] Hanneke, S., Karbasi, A., Mehrotra, A. and Velegkas, G. On union-closedness of language generation. NeurIPS, 2025.

---

> > ### Author Rebuttal · Reviewer_aCSo · 2026-04-03
> >
> > Thanks for the response! I think my issues are resolved apart from the presentation, which I still think is below bar. That being said, I will not object to the paper's acceptance based on my reading of other reviewer's comments + rebuttals.

---

> > > ### Author Response · Authors · 2026-04-04
> > >
> > > Thank you for the feedback on how to improve our manuscript. We will update the draft to incorporate the changes discussed above and the presentation-related feedback.

---

### Official Review · Reviewer_Bjz6 · 2026-03-13

**Soundness:** 3
**Presentation:** 3
**Significance:** 3
**Originality:** 3
**Overall Recommendation:** 5
**Confidence:** 1

**Summary:**

The paper studies the risk of model collapse when training an LLM on LLM's generated data. The paper provides an analysis from a learning theory perspective via a replay adversary. It shows that replay can limit learning under weaker generation while remaining benign under uniform assumptions

**Compliance With Llm Reviewing Policy:**

Affirmed.

**Final Justification:**

The paper is strong in originality, technical soundness, and the rebuttal clearly addressed my main concerns regarding the practicality of the framework.

**Key Questions For Authors:**

NA

**Strengths And Weaknesses:**

Strengths:

---

- The paper provides theoretical results that explain when practical mitigation strategies, such as data cleaning, watermarking, and output filtering, can succeed or fail.

- The paper provides fine-grained theoretical characterizations of when replayed synthetic data is useful vs harmful.

- The paper presents a principled analysis of how replayed machine-generated data affects future training.

---

Weaknesses:

---

- The analysis is conducted under a theoretical framework (language generation in the limit) that may rely on assumptions, which limit how the results can be translated into actionable methods.

- The replay adversary formulation may simplify how synthetic data actually re-enters training corpora, since real-world contamination from machine-generated content is likely to be noisy and indirect.

---

> ### Author Rebuttal · Authors · 2026-03-30
>
> We thank the reviewer for their feedback. We briefly reiterate what we consider our main contributions to be and then discuss the points raised by the reviewer.
>
> In this work, we provide the first learning-theoretic analysis of model collapse in generative models; we do so through the lens of the language generation framework. As noted by the reviewer, this framework makes some simplifying assumptions to make the problem of modeling generation more tractable. We agree that improving the framework to address these limitations is an important direction for the broader community. However, this is outside the scope of the present work.
>
> We also agree that our replay framework might not capture the entirety of ways artificial data could re-enter the training pipeline, and we will add this point to the discussion section. At the same time, our replay adversary provides a minimal and clean way to analyse the impact of recursive training on generative tasks: our positive results are obtained via algorithms that mimic heuristics used in practice while our impossibility results identify the regimes in which these methods can break down.
>
> We are happy to answer any questions that might have arisen in the meantime, and we refer the reviewer to our answers to other reviews (especially to reviewers **MjSZ** and **95KP**) where we further acknowledge and elaborate on the limitations of the language generation framework.

---

> > ### Author Rebuttal · Reviewer_Bjz6 · 2026-04-03
> >
> > Thanks for your reply regarding the weaknesses I have raised.

---

### Official Review · Reviewer_95KP · 2026-03-13

**Soundness:** 3
**Presentation:** 4
**Significance:** 4
**Originality:** 4
**Overall Recommendation:** 5
**Confidence:** 2

**Summary:**

The authors present a learning theoretic framework for understanding model collapse in the presence of "replay" data generated by the model itself in the training data stream.

**Compliance With Llm Reviewing Policy:**

Affirmed.

**Final Justification:**

The authors addressed my concerns.  However, due to my limited expertise in this area, I'm reluctant to raise my rating higher than accept (5).  It's a great paper, from my  perspective, and I think the topic is important, I'm not not a theorist so I don't feel qualified to rate it as strong accept.

**Key Questions For Authors:**

1. In the proof for theorem 6.2, you define the finite class of four hypotheses  supp((h_i)^-) and supp((h_i)^+) (i \elem {1,2}) to exclude 0.  Then, in the proof, you say "let x_1=0" and then x_1 belongs to the support of all hypotheses in H.  But based on the previous definitions, 0 is not an element of the support of any hypothesis in the class.  Since the adversary must play an example from the target hypothesis, it seems like the sequence can't begin with x_1=0?

2. Can you address the following limitations, and discuss how they affect the relevance and applicability of your paper to practical LLM collapse?
a. The generator in this case is assumed to be deterministic, but real transformer models are stochastic.
b. There's an assumption that a perfect oracle exists (in the Witness Protection part), but there's no oracle to determine if generated data matches the true distribution of human language.
c. In the replay mechanism, it's assumed that the adversary injects exact past outputs, but synthetic data can come from a wide range of sources, and may be modified from its original form.
d. Real world data collection is heuristic and stochastic, and not actively adversarial, so this work is mostly confined to a "worst case" scenario as opposed to an inevitable outcome.

Note: I'm not an expert in this particular area of ML, and I'm happy to modify my rating based on discussions with the author.

**Limitations:**

yes

**Strengths And Weaknesses:**

Strengths:  Model collapse due to the increasing amount of LLM generated data on the internet is a much discussed topic. Though I'm not an expert in this area, the proofs and arguments seem mostly sound.

Weaknesses:
There's one issue I noted in the proof of theorem 6.2.  I'll elaborate in the questions.

Other than that, I feel there are some simplifications that make this difficult to apply to practical models.
1. The generator in this case is assumed to be deterministic, but real transformer models are stochastic.
2. There's an assumption that a perfect oracle exists (in the Witness Protection part), but there's no oracle to determine if generated data matches the true distribution of human language.
3. In the replay mechanism, it's assumed that the adversary injects exact past outputs, but synthetic data can come from a wide range of sources, and may be modified from its original form.
4. Real world data collection is heuristic and stochastic, and not actively adversarial, so this work is mostly confined to a "worst case" scenario as opposed to an inevitable outcome.

---

> ### Author Rebuttal · Authors · 2026-03-30
>
> We thank the reviewer for the careful reading and the detailed questions.
>
> 1. **Proof of Theorem 6.2.** By $\mathbb{Z}_{\leq 0}$ we mean the set of all negative integers union 0 (i.e., 0,-1,-2,-3,...). We will specify this better in the revised version. We also note that we provided additional detail on the proof of Theorem 6.2 in the response to reviewer **aCSo** and that we are happy to provide any additional clarification needed.
>
> 2. **Limitations.** Our analysis focuses on isolating the fundamental effect of synthetic outputs re-entering future training data (rather than on reproducing the full complexity of current LLM training pipelines, which remains an open challenge for the whole community). The fact that this framework is agnostic to implementation details is both a limitation but also a strength: while it is harder to directly prescribe implementation-level feedback for practitioners, it allows us to identify fundamental possibility and impossibility boundaries that are not tied to any particular architecture or optimization procedure—our findings will hold if we move from current LLM architectures to different ones in the future. We now dive into the specific questions:
>
> a. **Deterministic generator.** Ignoring nondeterminism arising from complex deployment details (like floating point issues), after "fixing the seed", transformer-based generators are deterministic. Therefore, if the adversary knows the seed, our theory covers transformers as well. We agree, however, that extending the theory to truly randomized algorithms is an important direction for future work and something we have been considering actively. Nevertheless, this issue remains largely unaddressed by the language identification and generation literature.
>
> b. **Oracle access.** We clarify that we do not assume oracle access to the target language; that is, we do not assume that the generator can determine whether any given string belongs to the support of the target hypothesis. What we do assume is that for any element $x_j \in \mathcal{X}$ and any $h_i \in \mathcal{H}$ the algorithm can query whether $x_j \in supp(h_i)$, which is equivalent to being able to evaluate $h_i(x_j)$. This is a minimal assumption: when dealing with multiple languages, given a language's representation and a specific word, the algorithm can definitively access whether the word belongs to it. Crucially, however, because the index of the target language is unknown, the generator cannot directly query membership in the target language. This is standard in the language identification [Ang80] and generation [KM24] literature, and differs from the "feedback" setting in [CP25], where additional membership-query access to the target language is studied.
>
> c. **Replay model.** We agree that, while our framework  attempts to isolate the effect of replay in a clean and minimal way, it does not perfectly capture all of its intricacies. We will add these points raised by the reviewer to our discussion section. As a first step toward a more complex yet still tractable model, we note that our Theorem 5.2 can be combined with Lemma 4.3 from [MVYZ25] to show that there exists an algorithm that generates in the limit all countable hypothesis classes under replay *and* finite contaminations (i.e., arbitrarily noisy or omitted elements).
>
> d. **Adversarial setting.** Similarly, extending our framework from a worst-case adversarial setting to a stochastic one is an important direction for future work, and we will emphasize this more in the discussion. We agree with the reviewer: as already noted in the discussion, introducing a distribution over the data might help circumvent some of our strong impossibility results. However, the current setting has the advantage of making only a minimal modification to the standard framework of language generation in the limit, thus allowing for a direct comparison between what is generatable with and without replay.
>
> ---
>
> [Ang80] Angluin, Dana. Inductive inference of formal languages from positive data. Information and control, 1980.
>
> [CP25]  Charikar, Moses, and Chirag Pabbaraju. Exploring facets of language generation in the limit. COLT, 2025.
>
> [KM24] Kleinberg, Jon, and Sendhil Mullainathan. Language generation in the limit. NeurIPS, 2024.
>
> [MVYZ25] Mehrotra, Anay, Grigoris Velegkas, Xifan Yu, and Felix Zhou. Language Generation with Infinite Contamination. arXiv:2511.07417, 2025.

---

> > ### Author Rebuttal · Reviewer_95KP · 2026-03-31
> >
> > The authors have addressed my questions.  While I think that there still remain some uncertainty about how well this applies to practical systems, I agree with the authors that these types of theoretical bounds are still useful, and the work addresses an important, unaddressed problem in the literature.

---

### Official Review · Reviewer_MjSZ · 2026-03-14

**Soundness:** 3
**Presentation:** 3
**Significance:** 2
**Originality:** 3
**Overall Recommendation:** 5
**Confidence:** 2

**Summary:**

The paper studies the phenomenon of model collapse in LLM from a learning theory perspective. Model collapse refers to the degradation that can occur when models are trained on data that includes outputs from earlier models. The authors formalize this phenomenon using a theoretical framework for language generation in the limit, and introduce a replay adversary. This adversary injects previously generated outputs from the model back to the training data stream. The key contribution is to frame model collapse as a learning theory property of the training process rather than an empirical one.

**Compliance With Llm Reviewing Policy:**

Affirmed.

**Final Justification:**

The authors have clarified most of the concerns. I appreciate the follow-up clarification and have increased my score.

**Key Questions For Authors:**

1. How do the theoretical notions of uniform vs non-uniform generation correspond to the practical usage of LLMs?
2. Can the authors include some empirical simulations showing the replay induced degradation?
3. How would the theoretical analysis change or stay constant if the hypothesis class becomes large for e.g. models with billions of parameters?

**Limitations:**

1. Some empirical validation could have been more affirming.
2. Would it be stronger if the framework captures the features of LLM training like gradient based optimisation or token level prediction objectives?

**Strengths And Weaknesses:**

Soundness - The paper is technically sound and provided formal definitions of replay and generative language settings. What is a bit unclear is how well the theoretical notions of uniform vs non-uniform generation correspond to practical LLM behaviors.

Presentation - The paper is conceptually well motivated and well-written. Some examples or intuitive explanations might help bridge the gap between theory and practice.

Significance - The paper addresses an important issue of model collapse or distribution drift. It provides a strong formal theoretical approach to the problem. The impact may be increased by including some empirical experiments or simulations.

Originality - The paper is novel in its theoretical framing of the underlying motivation -- self training loops cause degradation.

---

> ### Author Rebuttal · Authors · 2026-03-30
>
> We thank the reviewer for their feedback and answer their questions below.
>
> **Q1: Uniform vs non-uniform generation and LLMs.** Intuitively, these notions differ in what can be said about the size of the training corpus necessary to guarantee hallucination-free generation. In the *uniform* setting, there is a single corpus size (in principle, known a priori) that works for every target language/hypothesis within a family of languages (e.g., one size that works for learning all programming languages, another that works for all kinds of distinct reasoning tasks, and another that works for different styles of chatbot behaviors). In contrast, in the *non-uniform* setting, each target language (even within the same family) has its own threshold; these thresholds may vary substantially depending on the language being learnt (e.g., learning C might have a larger threshold than learning BASIC) but do not depend on the specific input stream of examples. In other words, given a language, any corpus of the right size will suffice. Finally, generation *in the limit* is weaker still: it is only known that the generator will eventually produce outputs that are hallucination-free after a finite time, but this finite time may be arbitrarily large depending on the specific target language and on the specific input stream. We believe all three notions deserve attention, just as in the supervised learning literature one distinguishes, in a similar way, between PAC learnability, non-uniform PAC learnability, and consistency [SB14].
>
> **Q2/L1: Experiments.** We agree that this phenomenon is also interesting to observe experimentally. In fact, we were motivated by several prior works that have already shown this behavior to be pervasive in recursive LLM training (under the notion of model collapse). Our contribution is *complementary*: we offer a rigorous theoretical perspective by establishing a clear separation between what is learnable and what is not learnable under a clean abstract learning-theoretic model of recursive training.
>
> **Q3: Size of the hypothesis class.** Our results depend on the complexity of the induced hypothesis class, not directly on the  parameter count. This is already shown in Table 1. We note that a fixed architecture with finite precision induces a *finite* class; idealized real-valued weights can induce an *uncountable* class; and *countable* classes form a natural intermediate regime (e.g., computable hypotheses). Our results show that replay sensitivity depends on the class size in a nontrivial way.
>
> **L2: Regarding gradient-based optimization and token-level objectives**, we agree that explicitly modeling these optimization dynamics would bring the theory closer to modern LLM training, and we think this would be an interesting direction for future work. Our goal here is narrower: to isolate the effect of replay on the task of generation itself, and it is not immediately clear how to incorporate optimization dynamics into this framework. Nevertheless, we will sharpen our discussion to make clear that this abstraction is both a strength (in that it identifies effects that are independent of implementation details such as architecture or training procedure) and a limitation (in that it does not model optimization dynamics or token-level losses directly), thereby suggesting several interesting directions for future work.
>
> ---
>
> [SB14]  Shalev-Shwartz, Shai, and Shai Ben-David. Understanding machine learning: From theory to algorithms. Cambridge university press, 2014.

---

> > ### Author Rebuttal · Reviewer_MjSZ · 2026-04-01
> >
> > The authors clarified most of the concerns. On the point of how uniform vs non-uniform generation correspond to practical usage of LLMs, it is a bit unclear as to what is the analogous example in the setting where we use pre-training or fine-tuning data to train LLMs?

---

> > > ### Author Response · Authors · 2026-04-03
> > >
> > > Thanks for the engaging discussion! We are glad to hear we resolved most of your concerns and happy to elaborate on the remaining one.
> > >
> > > One of the crucial aspects of the language generation framework is that the generator is learning exclusively from *positive* examples, that is, documents that are part of the target language/desired behavior (as opposed to also being shown negative examples such as incorrect, malformed, or prohibited text). Therefore, our theory covers *both* LLM pre-training and instruction tuning/supervised fine-tuning. However, it does not explicitly distinguish between these two cases, since it is a general theory that does not capture weight initialization. By contrast, our theory does not directly apply to preference optimization or RLHF, where the LLM is provided with feedback in the form of positive *and* negative examples. Nonetheless, there are extensions to the language generation framework that capture these forms of post-training [CP25], and follow-up work could explore whether additional access to such negative feedback might help overcome some of the challenges of the replay setting.
> > >
> > > Loosely speaking, if we take the hypothesis class $\mathcal{H}$ to be the family of all natural languages, we can view *pre-training* as learning one—or, more realistically, some—of these languages. As an example in the context of *fine-tuning*, we can instead take $\mathcal{H}$ to be a family of desired behaviors, such as a chatbot, a tool-using agent, or an advanced reasoning agent. In this analogy, *uniform* generatability would imply that we would know an a priori bound on the training time required to achieve hallucination-free generation for any training corpus and any language (for pre-training) or behavior (for fine-tuning). Although this guarantee is appealing in practice and easier to analyze theoretically, the *non-uniform* guarantee is likely more realistic for LLMs in practice given that some languages/behaviors might be considerably more difficult to learn than others. Additionally, since we consider worst-case adversarial example sequences $(x_t)_{t\geq 1}$ (thus corresponding to the worst possible ordering of documents in a potentially infinite training corpus), generatability *in the limit* is likely to be the more fitting setting overall, and much of the analysis in our paper is carried out in this setting.
> > >
> > > Finally, the reviewer’s question naturally leads to an interesting direction that we (or the field more broadly) have not yet considered, namely how to explicitly model fine-tuning in this framework, and what implications this has for the generatability guarantees. One possible approach is to view fine-tuning as starting from a good initial guess for the language and then exploring only a restricted subset of the original hypothesis class, since the limited number of fine-tuning steps restricts the set of languages that can be reached. Under such a model, it seems plausible that stronger guarantees could be obtained for this smaller class than for the original one; for example, one might obtain a uniform guarantee even when the original class admits only a non-uniform one.
> > >
> > > ---
> > >
> > > [CP25]  Charikar, Moses, and Chirag Pabbaraju. Exploring facets of language generation in the limit. COLT, 2025.

---

### Decision · Program_Chairs · 2026-04-30

**Decision:**

Accept (regular)

**Comment:**

This paper makes a clear theoretical contribution on an emerging topic: model collapse under replayed synthetic data. Reviewers found the replay framework natural and the results technically meaningful, especially the fine-grained characterization of when replay is benign versus harmful.

The rebuttal addressed the main concerns by clarifying the proof details, the intended connection to practical LLM settings, and the limitations of the abstract framework. The remaining reservations were mostly about presentation and abstraction rather than correctness or significance. I would ask the author to revise the final version based on the discussion and make the paper more self-contained.

I recommend acceptance.